# Isolation, engineering and ecology of temperate phages from the human gut

Sofia Dahlman[1], Laura Avellaneda-Franco[1], Emily L. Rutten[2,3], Emily L. Gulliver[2,3], Sean Solari[2,3], Michelle Chonwerawong[2,3], Ciaren Kett[1], Dinesh Subedi[1,4,5], Remy B. Young[2,3], Nathan Campbell[1], Jodee A. Gould[2,3], Jasmine D. Bell[2,3], Callum A. H. Docherty[2,3], Christopher J. R. Turkington[6], Neda Nezam-Abadi[6], Juris A. Grasis[7], Dena Lyras[8], Robert A. Edwards[9], Samuel C. Forster[2,3,10] & Jeremy J. Barr[1,10] ✉

Large-scale metagenomic and data-mining efforts have revealed an expansive diversity of bacteriophages (phages) within the human gut[1–3]. However, functional understanding of phage–host interactions within this complex environment is limited, largely due to a lack of cultured isolates available for experimental validation. Here we characterize 134 inducible prophages originating from 252 human gut bacterial isolates using 10 different induction conditions to expand the experimentally validated temperate phage–host pairs originating from the human gut. Importantly, only 18% of computationally predicted prophages could be induced in pure cultures. Moreover, we construct a 78-member synthetic microbiome that, when co-cultured in the presence of human colonic cells (Caco2), led to the induction of 35% phage species. Using cultured isolates, we demonstrate that human host-associated cellular products may act as induction agents, providing a possible link between gastrointestinal cell lysis and temperate phage populations[4,5]. We provide key insights into prophage diversity and genetics, including a genetic pathway for domestication, finding that polylysogeny was common and resulted in coordinated prophage induction, and that differential induction can be influenced by divergent prophage integration sites. More broadly, our study highlights the importance of culture-based techniques, alongside experimental validation, genomics and computational prediction, to understand the biology and function of temperate phages in the human gut microbiome. These culture-based approaches will enable applications across synthetic biology, biotechnology and microbiome fields.

The human gut microbiota consists of a plethora of microorganisms along with the viruses that infect them, including phages. These viruses are thought to shape the gut microbial community through predation, horizontal gene transfer and lysogenic conversion[6,7]. Recent advances in computational mining of gut metagenomes have revealed an expansive collection of viral metagenome-assembled genomes and efforts cataloguing this diversity have led to the discovery of several important viral families[1–3,8–10]. Moreover, lysogeny is common within the gut, with up to 90% of bacteria predicted to harbour prophages[11,12]. However, the extent to which these prophages re-enter lytic replication remains unclear. For example, the inactivation of resident prophages represents a common strategy whereby the bacterial population can escape lysis while maintaining beneficial phage genes[13,14].

Furthermore, initiation of lytic replication by resident prophages is complex, involving both host- and phage-specific cues[15,16]. Within the gut, little is known about temperate phages and how they interact with our commensals.

## Induction of human gut isolates

Advances in cultivation of the microbiota have enabled the isolation and archiving of previously 'unculturable' gut bacterial species[17,18], along with their phages. Here we use a collection of 252 human gut bacterial isolates (50 Actinomycetota, 1 Fusobacteriota, 51 Bacillota, 57 Pseudomonadota and 93 Bacteroidota) to computationally identify and experimentally validate inducible prophages (Fig. 1a and

[1]School of Biological Sciences, Monash University, Melbourne, Victoria, Australia. [2]Centre for Innate Immunity and Infectious Disease, Hudson Institute of Medical Research, Melbourne, Victoria, Australia. [3]Department of Molecular and Translational Sciences, Monash University, Melbourne, Victoria, Australia. [4]Centre to Impact AMR, Monash University, Melbourne, Victoria, Australia. [5]School of Optometry and Vision Science, UNSW Medicine, University of New South Wales, Sydney, New South Wales, Australia. [6]APC Microbiome Ireland & School of Microbiology, University College Cork, Cork, Ireland. [7]Department of Molecular and Cell Biology, University of California, Merced, CA, USA. [8]Monash Biomedicine Discovery Institute Department of Microbiology, Monash University, Melbourne, Victoria, Australia. [9]College of Science and Engineering, Flinders University, Adelaide, South Australia, Australia. [10]These authors contributed equally: Samuel C. Forster, Jeremy J. Barr. ✉e-mail: Jeremy.barr@monash.edu

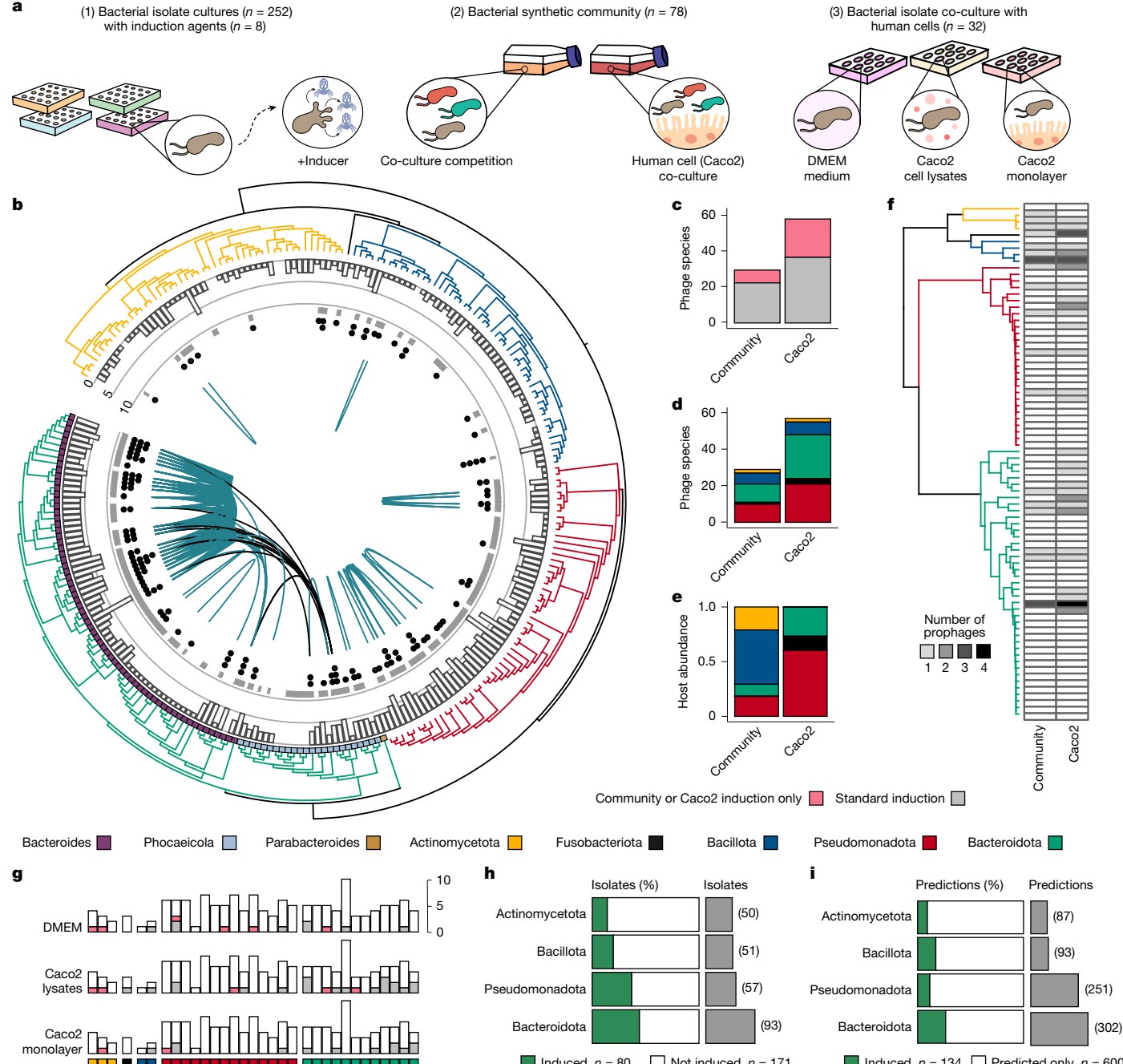

**Fig. 1 | Gut prophages induced in pure and synthetic bacterial community cultures. a**, Schematic of the methods used. (1) Induction of pure-culture bacterial isolates with standard induction agents. (2) Bacterial synthetic community co-cultured with and without Caco2 cell monolayer. (3) Induction of bacterial isolate pure cultures with cell culture medium, Caco2 cells or cell products. **b**, Phylogenetic tree of isolates. Actinomycetota is shown in yellow (50), Fusobacteriota in black (1), Bacillota in blue (51), Pseudomonadota in red (57) and Bacteroidota in teal (93). The outer ring shows genera of Bacteroidota; the white bars show high-quality (> 50% complete) predictions; the inner ring shows sequenced isolates (grey); and the dots represent induced prophages (black). The green lines connect isolates sharing induced prophage species and the black lines connect isolates of different genera with the same prophage species. **c**, The number of induced prophage species within the community (left; *n* = 29) and community co-cultured with Caco2 cell monolayer (right; *n* = 57); the grey shading indicates prophage species that were previously

identified in standard conditions and pink shading shows prophage species induced only within the synthetic community. **d**, The number of induced prophage species as in **c**, coloured by phyla. **e**, Host abundance within the synthetic microbiome community at the phyla level, shown as the average of timepoints (*n* = 3) and replicates (*n* = 5). **f**, Phylogenetic tree of the bacterial synthetic community (*n* = 78), and heat map depicting induced prophages within the community (left; *n* = 22) and the community co-cultured with Caco2 cell monolayer (right; *n* = 42) identified using KrakenUnique. **g**, Isolates (*n* = 32) picked for pure culture induction in DMEM cell medium, Caco2 lysates and Caco2 cell monolayer. The bars represent the number of high-quality predictions (146); white shows total predictions, pink shows newly induced prophages (*n* = 9), grey squares show prophages also induced under standard conditions (*n* = 20). **h**, The percentage of induced isolates and the total number of isolates. **i**, The percentage of induced and the total number of high-quality prophage predictions. Fusobacteriota was excluded from **h** and **i** due to a single isolate.

Supplementary Table 1). We began by exposing our bacterial isolate cultures to eight different induction agents and conditions, which included a standard medium control, well-known inducing agents such as mitomycin C (0.3 and 3 µg ml[−1]) and hydrogen peroxide (0.5 mM), along with lesser-known induction conditions with potential relevance to the gut, including the sugar substitute Stevia (3.7 and 37 mg ml[−1]) and two starvation conditions (50% carbon depletion and 100% short-chain fatty acid (SCFA) depletion)[19–23]. After induction, the samples were processed for DNA extraction and 433 viral induction samples that passed our inclusion criteria were sequenced (Extended Data Fig. 1a,b and Supplementary Table 2). This resulted in the detection of 125 inducible gut prophages, representing 63 (23%) phage species (95% average nucleotide identity (ANI), over 85% alignment fraction (AF))[24] from 73 (29%) bacterial isolates (5 Actinomycetota, 1 Fusobacteriota, 10 Bacillota, 17 Pseudomonadota and 40 Bacteroidota) (Fig. 1b).

## Human cellular products induce prophages

To further expand on the prophage induction triggers tested above, we constructed a synthetic bacterial community based on a subset of our isolates ($n = 78$, 4 Actinomycetota, 1 Fusobacteriota, 4 Bacillota, 28 Pseudomonadota and 41 Bacteroidota; Supplementary Table 1). Using this community, we investigated the effects of bacterial co-culture, in which the competition for resources, production of microbial byproducts and quorum sensing may affect prophage induction, as well as community co-culture with a monolayer of human colonic epithelial cells (Caco2), to investigate human host-associated factors[5,16] (Extended Data Fig. 1c). In total, 29 phage species out of 162 (17%) were identified as induced in community co-culture using read mapping, yet, notably, 57 phage species (35%) were induced within the Caco2 co-culture, with a total of 22 phage species being newly identified as inducible across both experiments (Fig. 1c). However, there was a shift in the bacterial community composition within the Caco2 co-culture, dominated by Pseudomonadota and Bacteroidota, potentially leading to detection of prophages induced from isolates extinct in the community only co-culture[25] (Fig. 1d,e). In a complementary approach, we detected the induction of distinct prophage–host pairs by identifying unique $k$-mers within each prophage genome. Out of the 338 predicted prophages within the community, 150 contained unique $k$-mers, allowing for the detection of 43 (29%) induced prophages, 21 of which were detected only in the Caco2 co-culture (Fig. 1f).

Considering the increased prophage induction within the Caco2 co-cultured community, we wanted to investigate whether human cells or cell lysis products (independent of community effects) act as prophage induction triggers. To this end, we selected 32 bacterial isolates from the community (3 Actinomycetota, 1 Fusobacteriota, 2 Bacillota, 14 Pseudomonadota and 12 Bacteroidota; Supplementary Table 1) for pure culture induction assays using Caco2 cell monolayers, Caco2 cellular lysates and DMEM cell culture medium alone (Extended Data Fig. 1a). These conditions induced 29 prophages, with 25 observed within the Caco2 cell lysate condition and 14 in the Caco2 monolayer and DMEM cell medium (Fig. 1g). Importantly, nine of the induced prophages had not been previously detected in our bacterial isolate cultures using standard induction agents, indicating that human host-associated cellular products act as induction triggers. Taken together, 35 out of 146 (24%) prophages were found to be inducible across all conditions within these 32 bacterial isolates.

## Only a fraction of gut prophages were induced

Consistent with previous reports of substantial lysogeny within the human gut[11], 237 out of 252 of bacterial isolates (94%) were computationally predicted to contain high-quality prophage regions. However, across all 10 of our tested induction conditions, only 32% (80 out of 252) of isolates were induced and 18% (134 out of 736) of the high-quality prophage predictions, or 24% (68 out of 274) of high-quality prophage species, corresponded to experimentally inducible prophages in pure culture conditions (Fig. 1h,i and Supplementary Table 3). The highest concordance between inducible and predicted prophage regions was observed within Bacteroidota isolates, in which 80 predictions (27%) from 41 isolates (44%) were inducible. Comparatively, in Pseudomonadota, which had the highest number of predicted prophages (4.5 per isolate), just 12% of prophages were found to be inducible. Moreover, combining the synthetic communities, a total of 36% of prophage species was detected as induced, coinciding with recent reports from human gut metagenomes (8–36%)[26,27]. Although our experimental approach does not provide comprehensive identification of all gut prophages due to factors including detection limits and potential unidentified induction conditions, it is likely that a substantial portion of predicted prophages within our dataset rarely undergo induction.

## Taxonomy of induced gut temperate phage

We next looked to assign phage taxonomy to our induced temperate phage collection using a database comprising 9,920 phage reference genomes. Given the inherent challenges in assigning taxonomy to phages[28], we applied both a gene voting-based search and a gene sharing network method using vContact2 (refs. 2,29) (Fig. 2a, Extended Data Fig. 3a and Supplementary Table 4) with taxonomy assigned to the highest taxonomic resolution shared between methods. The resulting classification assigned 133 phages to the *Caudoviricetes* order and one phage within the *Faserviricetes* order (Supplementary Table 3). In total, 26% (35 out of 134) of phages could be assigned to ICTV (International Committee on Taxonomy of Viruses) accepted taxa at the family level or lower. These belonged to previously reported phage taxa infecting Pseudomonadota (*Bcepmuvirus*, *Punavirus*, *Uetakevirus* and *Peduoviridae*), one *Spbetavirus* infecting Bacillota and 16 prophages belonging to the *Winoviridae* family infecting Bacteroidota. Although lacking ICTV classification, 30 genomes could be grouped into viral clusters (genus-subfamily level) together with previously described phages. Notably, 19 of these clustered with Hankyphage, a recently described virus thought to lysogenize several *Bacteroides* species[30]. Further taxonomic classification grouped ten prophages at the species level with Hankyphage, whereas the remaining nine clustered into seven potential novel species, forming a putative novel genus that we name Hankyvirus after the original phage characterized (Extended Data Fig. 2a). Comparing the Hankyvirus species to bacterial genomes in NCBI RefSeq database (95% ANI over 85% AF), we identified 52 host species originating from 9 genera and 5 families, indicating a broad host range of this genus (Extended Data Fig. 2b). Correspondingly, we find two Hankyvirus species induced within both *Bacteroides* and *Phocaeicola* isolates within our collection, providing experimental validation of these phages as actively replicating across these two host genera.

## Inducible temperate phages are prevalent

We next sought to place our temperate phage genomes within the larger context of the human gut by comparing their prevalence to the reference genomes, including the *Crassvirales* order[31]. Approximately half of our inducible prophage species (28 out of 68) could be detected in gut viromes[9] ($n = 1,241$; Fig. 2b and Supplementary Table 5). LoVEphage, a recently discovered Bacteroidota phage[9,10], was the most common, being detected in around 8% (97 out of 1,241) of the viromes and representing up to 64% of reads within one virome (Supplementary Table 3). Comparatively, the most abundant *Crassvirales* genome, belonging to the alpha/gamma family, was found in approximately 19% of the viromes investigated (Extended Data Fig. 3b). Three phages in our collection were species-level members of LoVEphage, induced from *Bacteroides thetaiotaomicron*, *Phocaeicola dorei* and *Phocaeicola vulgatus* hosts (Extended Data Fig. 2b). An additional eight phage

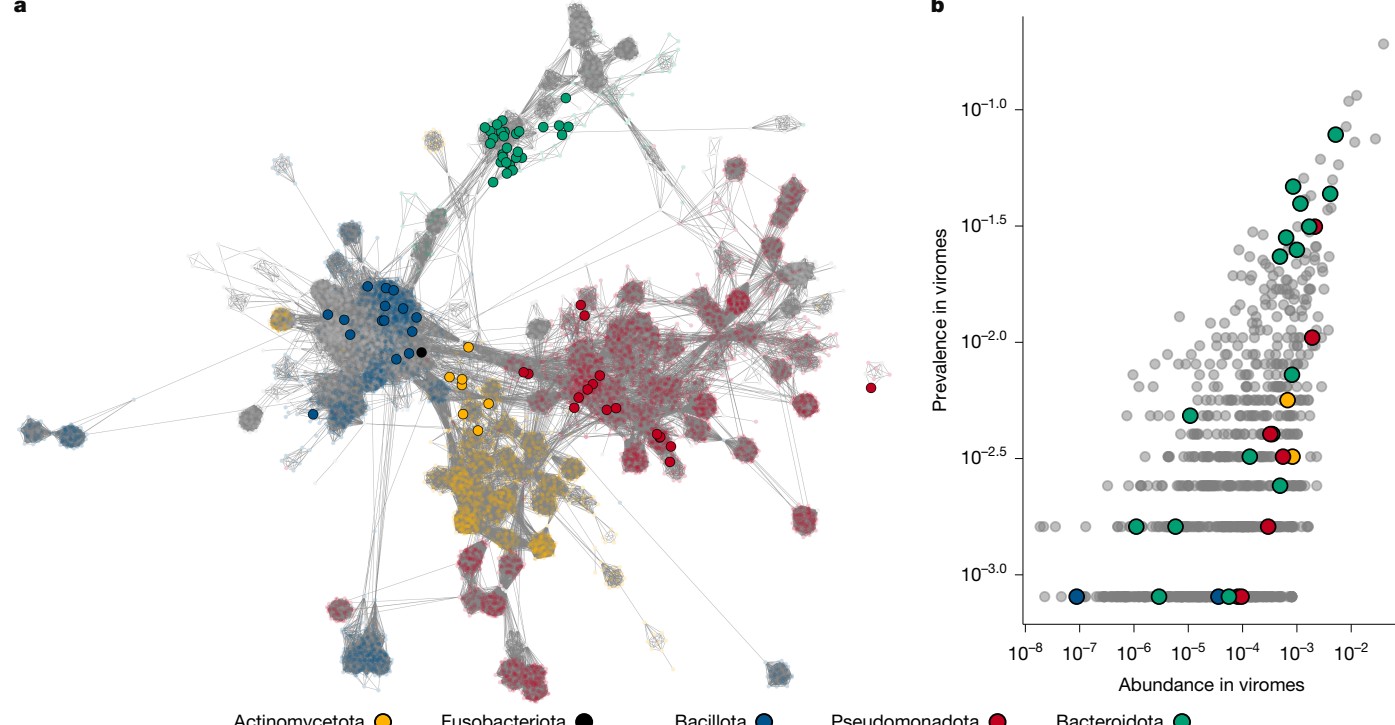

**Fig. 2 | Taxonomy and prevalence of induced temperate phages within gut viromes. a**, Gene sharing network of inducible prophage species (solid circles; n = 68) coloured by host. Actinomycetota is shown in yellow (7), Fusobacteriota in black (1), Bacillota in blue (15), Pseudomonadota in red (17) and Bacteroidota in teal (28). Database representatives (9,920) are translucent and coloured by host phyla when applicable, and are otherwise coloured grey. The portion of the network connecting to induced prophages is shown; the full network is shown in Extended Data Fig. 3. **b**, The mean fractional abundance and detection frequency (prevalence) of *Caudoviricetes* phages within 1,241 viromes originating from the human gut. A minimum of 70% coverage over the phage was required to be counted as present within a virome. The bacterial host phyla of inducible prophage species coloured as in **a**, and database reference genomes (n = 875) are shown in grey.

species were detected in 2–5% of gut viromes (Supplementary Table 3). These included the four species within the Hankyvirus genus, one *Uetakevirus* infecting *Escherichia coli* and three previously uncharacterized Bacteroidota phages (Wilby, Saffi and Shia; Extended Data Fig. 3c).

## DGRs are common within gut prophages

Discernible integrase or site-specific recombination genes, both of which are used as hallmark genes for a temperate lifestyle[32,33], were absent in 28% (19 out of 68) of our inducible phage species, including Hankyviruses. We found transposases in ten of these viruses, while the remaining nine lacked any discernible integration genes, illustrating the difficulty in assigning phage lifestyle based on genomic data alone. Diversity-generating retroelements (DGRs) are prevalent within the gut virome, and tail-targeting DGRs are known to enable rapid host switching in a *Bordetella* phage[34,35]. We found DGRs in 19% (13 out of 68) of our inducible prophage species, the majority of which were seen in Bacteroidota phages, in which 43% (12 out of 28) of species encoded DGRs targeting known and genomically predicted tail proteins (Supplementary Table 3). Concordantly, we found eight Bacteroidota prophage species actively replicating across different bacterial species, three of which replicating across different bacterial genera (Fig. 1b (connecting lines)), highlighting the involvement of DGRs in phage host range expansion through diversification of tail proteins[34]. More recently, bacterial DGRs were implicated in anti-phage defence mechanisms, and targeted engineering using DGRs accelerated evolution within both host receptor and the reciprocal phage binding domain[36,37]. Notably, we found four prophage species containing DGRs that encoded a second variable region (VR) targeting genes distal from the reverse transcriptase cassette[9] (Extended Data Fig. 3c). The second VR was found in proximity to counter defence genes, such as DNA methyltransferase, indicating further involvement of DGRs in the phage–host arms race[34].

## Differential gene enrichment patterns

The retention of cryptic prophages is known to provide the host with adaptive fitness advantages and has been shown to result in a bimodal size distribution of prophage genomes[13,14]. Concordantly, we find bimodal length distributions of prophages across all host phyla within our collection (Extended Data Fig. 4a), with the early peak corresponding to sequences with low completeness scores (<50% complete, n = 1,236) and later peaks corresponding to high-quality predictions (>50% complete, n = 736) and experimentally inducible prophages (n = 134; Fig. 3a and Supplementary Table 6). To investigate differences in gene content between these groups, we performed gene enrichment analysis of annotated PHROG gene categories and found that small prophage genomes lacked essential phage genes (such as structural, head and packaging, and lysis genes) but were enriched in accessory genes and genes of unknown function[38] (Extended Data Fig. 4b). Similarly, when limiting the comparison to high-quality predictions, we found an enrichment of structural genes (head and packaging, connector, lysis and tail) within induced prophages, whereas non-induced predictions were enriched for accessory genes and genes of unknown function, indicating that a subset of high-quality predictions might be cryptic prophage-like elements or poor predictions (Fig. 3b).

We next sought to investigate potential genetic mechanisms leading to trapping of prophages within the host genome by comparing experimentally induced prophages to highly similar, non-induced prophages (95% ANI, 85% AF; Supplementary Table 7). To classify these non-induced prophages as putatively cryptic, we restricted the analysis to prophages

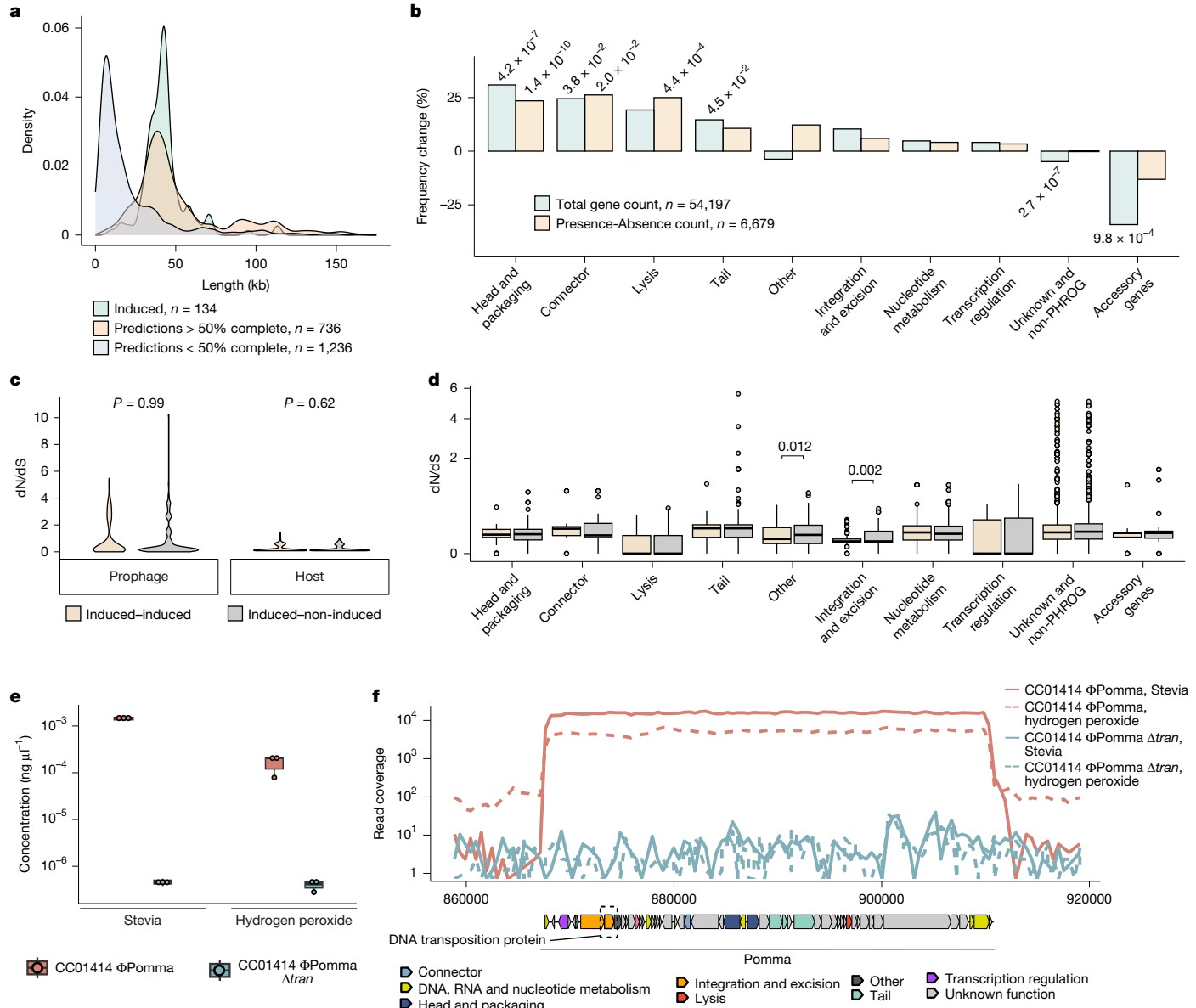

**Fig. 3 | Comparison of induced versus predicted prophages. a**, The length distribution of induced, high-quality (>50% complete) and low-quality (<50% complete) predicted prophage genomes. **b**, The percentage frequency change in PHROG gene categories between induced and high-quality (>50% complete) predicted prophage genomes, counting the total number of genes (blue) or the presence/absence count of each gene category within prophage genomes (yellow). Significant *P* values (shown above the bars) were calculated using two-sided Fisher's exact tests and adjusted using the Hochberg method. **c**, The dN/dS ratio between induced–induced or induced–non-induced prophage pairs (left; *n* = 313) and between their hosts (right; *n* = 389). No significant difference was found based on two-sided Wilcoxon rank-sum tests. **d**, The dN/dS ratio of PHROG gene categories (*n* = 7,893) between induced–induced and induced–non-induced prophage pairs. Significant *P* values (shown above the brackets) were calculated using two-sided Wilcoxon rank-sum tests and adjusted

using the Hochberg method. The box plots show the median (centre line), first and third quartiles (box limits), and the whiskers extend to ±1.5× the interquartile range; outliers are shown as dots. **e**, Absolute qPCR quantification of ΦPomma in Stevia (left, *n* = 3 biological replicates) and hydrogen peroxide (right, *n* = 3 biological replicates). The ΦPomma wild type is shown in red and ΦPomma Δ*tran* is shown in blue. The box plots show the median (centre line), first and third quartiles (box limits), and the whiskers extend to ±1.5× the interquartile range; the means of three technical replicates are shown as dots. **f**, DNA-sequencing reads from CC01414 ΦPomma wild type and ΦPomma Δ*tran* mapped to the *Bacteroides faecis* CC01414 genome; colours are as in **e**. The Stevia condition is shown by the solid lines and the hydrogen peroxide condition is shown by the dotted lines. For the ΦPomma genome map, genes are coloured by PHROG categories; unknown genes are shown in grey. The deleted 'DNA transposition protein' gene is highlighted by a black dashed box.

that had been sequenced (but not induced) in the same condition(s) as their inducible counterparts, with the rationale that highly similar prophages should respond to the same induction triggers. This resulted in a total of 231 prophage pairs between 65 induced and 58 non-induced prophages. No significant changes were found in gene frequency (*P* > 0.05, Fisher's exact test), indicating that, although gene loss may be characteristic of cryptic prophages, it is unlikely to be the initial cause of inactivation. Moreover, although we detected 201 homologous

gene transfer (HGT) and 65 insertion–deletion events, there was no significant difference in the number of total events when compared to a set of high sequence similarity induced prophage pairs (222 pairs, *P* = 0.46, Wilcoxon rank-sum test). Comparing host ANI between the induced and non-induced prophage pairs (Extended Data Fig. 4c,d), we found no association with induction (*P* = 0.6, Pearson's correlation), suggesting that prophage inactivation was not driven by diversification of the host or integration into divergent non-permissive hosts.

## Excision gene mutations trap prophages

To investigate whether non-induced prophages have an elevated number of mutations, we measured the ratio of non-synonymous to synonymous substitution rates (dN/dS) within the set of induced to non-induced prophage pairs, and their associated hosts. We found an overall elevated mutation rate in prophages (mean = 0.89, median = 0.18) compared with the host genome (mean = 0.16, median = 0.095, $P < 2 \times 10^{-16}$, Wilcoxon rank-sum test), but no significant difference between induced to induced or induced to non-induced prophage pairs or their host genomes ($P = 0.99$ and $P = 0.62$ respectively, Wilcoxon rank-sum test; Fig. 3c and Supplementary Table 8). Comparing gene substitution rates, we found 143 genes with elevated dN/dS rates (>1), indicating diversifying selection (Fig. 3d and Supplementary Table 9). Notably, the majority (110 out of 143) of these genes lacked a known function and 40% (57 out of 143) were associated with DGRs, highlighting an active and not yet deciphered role of DGRs within gut prophages. Importantly, within the non-induced prophages we found a significant increase in the dN/dS substitution rates in genes involved in integration and excision ($P = 0.002$, Wilcoxon rank-sum test), suggesting that non-functional mutations in these genes provides a pathway for the inactivation of prophages.

To test whether the inactivation of integration and excision genes could trap a prophage inside its host genome, we constructed a gene deletion mutant of the inducible prophage Pomma by knocking out its DNA transposition protein ($\Phi$Pomma $\Delta tran$) within the *Bacteroides faecis* host strain CC01414 (Extended Data Fig. 4e and Supplementary Table 10). From our previous inductions, we found that prophage Pomma was selectively induced by hydrogen peroxide and Stevia (37 mg ml$^{-1}$). Using these two inducers, we compared the induction of wild-type $\Phi$Pomma versus the $\Delta tran$ mutant using quantitative PCR (qPCR) and sequencing of chloroform and DNase-treated supernatants. qPCR analysis showed a 3.5 and 2.6 log increase in $\Phi$Pomma concentration within the wild type versus $\Delta tran$ deletion mutant in samples treated with Stevia and hydrogen peroxide, respectively (Fig. 3e and Supplementary Table 11). Through sequencing, we observed clear induction in the wild-type strain, with 35- and 17-fold increased coverage over the bacterial background in the Stevia-treated and hydrogen-peroxide-treated samples, respectively, whereas no increase over the bacterial background was detected in the $\Delta tran$ mutant strain (Fig. 3f).

## Phyla-specific cues may govern induction

To determine whether prophage induction was linked to phylogeny, we examined the induction response across our ten conditions and standard growth control (Fig. 4a and Supplementary Table 3). Combined, the two concentrations of mitomycin C induced the largest number of prophages ($n = 70$) and the most Pseudomonadota prophages ($n = 17$). Hydrogen peroxide induced 43 prophages, including the largest number of Bacteroidota prophages ($n = 35$). However, these well-known induction agents exhibited only marginally increased induction compared to spontaneous induction during standard growth condition ($n = 36$). The Caco2 human cell induction conditions induced 29 prophages from 32 tested hosts, with Bacteroidota ($n = 16$) followed by Pseudomonadota ($n = 9$) showing the largest numbers of induced prophages. Considerable overlap was observed between prophage induction in standard media and induction agents (mitomycin C, $n = 25$; hydrogen peroxide, $n = 15$; Stevia, $n = 19$; carbon depletion, $n = 9$; SCFA depletion, $n = 11$; Caco2 induction conditions, $n = 5$). Comparing induction conditions across each phylum, the only significant difference observed was within the Pseudomonadota phyla in response to 3 μg ml$^{-1}$ mitomycin C ($P = 0.024$, Fisher's exact test).

## Polylysogeny and host genetics influence induction

We next investigated polylysogeny across our collection and its influence on induction (Fig. 4b). Polylysogeny was most prevalent within the Bacteroidota isolates, in which 28 out of 41 (68%) of lysogens had more than one inducible prophage compared with 11 out of 38 isolates (29%) across the other phyla ($P = 0.002$, Fisher's exact test). We then compared whether polylysogeny influenced induction, observing a positive correlation between the number of co-inhabiting inducible prophages and conditions leading to induction of each prophage ($\tau = 0.22$, $P = 0.002$, Kendall's rank correlation; Fig. 4c). Prophages residing in polylysogens ($n = 90$) were induced on average in 2.7 conditions compared with 2.1 conditions in single lysogens ($n = 35$, $P = 0.03$, Wilcoxon rank-sum test), suggesting that polylysogeny may promote simultaneous prophage induction and reduce stability within lysogens.

Finally, we investigated differential induction within polylysogens by measuring the abundance of phage DNA in the supernatants of five highly similar (99% ANI) *Bacteroidota caccae* isolates harbouring the same two prophages ($\Phi$Wilby and $\Phi$Pomma; Supplementary Table 11). We identified an overall preferential induction of $\Phi$Wilby within standard medium ($P = 0.006$), but not in hydrogen-peroxide-treated samples ($P = 0.9$, Wilcoxon rank-sum test), with isolate CC01407 demonstrating the most marked difference between the two phages ($P = 0.026$, paired *t*-test; Fig. 4d). Calculating the ratio of $\Phi$Wilby over $\Phi$Pomma within each isolate, we found a significant variance of means between the isolates in both standard medium ($P = 0.012$) and hydrogen peroxide ($P = 0.0008$, analysis of variance (ANOVA)). These results implied that the host genetic background, even within highly similar isolates, may affect prophage induction. We previously identified phage $\Phi$Pomma as a transposable prophage that does not use site-specific integration, but randomly inserts into the host genome[39]. To investigate the prophage integration sites within our isolates, we used long-read sequencing on the five *B. caccae* strains. Genomic analysis identified $\Phi$Wilby integrated into the same tRNA gene location, which is characteristic of site-specific integration; however, the transposable prophage $\Phi$Pomma was found in four different genomic locations within the five isolates (Fig. 4e), implicating the integration site as a possible driver for the observed differential induction within these isolates.

## Discussion

The high microbial load within the human gut represents an optimal environment for temperate phages, as frequent interactions with their hosts provide ample opportunity for lysogeny[12]. Concordantly, the majority of bacteria within the gut are predicted to be lysogens, with up to 90% of bacteria harbouring at least one prophage[11,12]. However, the degree to which these prophages engage in lytic replication is poorly understood. Using our defined culture collection of 252 gut bacterial isolates, we predict that the majority harbour prophage-like elements (94%), but find that only a fraction of predicted prophages could be experimentally induced in pure culture (18%). Caveats to our approach include experimental cut-offs for detection and minimum amounts of DNA required for sequencing, which could exclude detection of low-level-inducing prophages. Moreover, there are probably biases towards induction and detection of *Caudoviricetes* prophages and, indeed, all but one prophage (from the *Inoviridae* family) belonged to this class. Moreover, considering little is known about prophage induction triggers within the gut, it is plausible that some of our isolates carry prophages that were not induced due to a lack of appropriate induction triggers. To address this, we constructed a synthetic microbiome community and co-cultured it with or without human cells to simulate the biologically relevant conditions within the human gut. Within the community co-culture, we estimate that around 29% of prophages were induced, with around half of these induced only in co-culture with human epithelial Caco2 cells. However, whether this induction

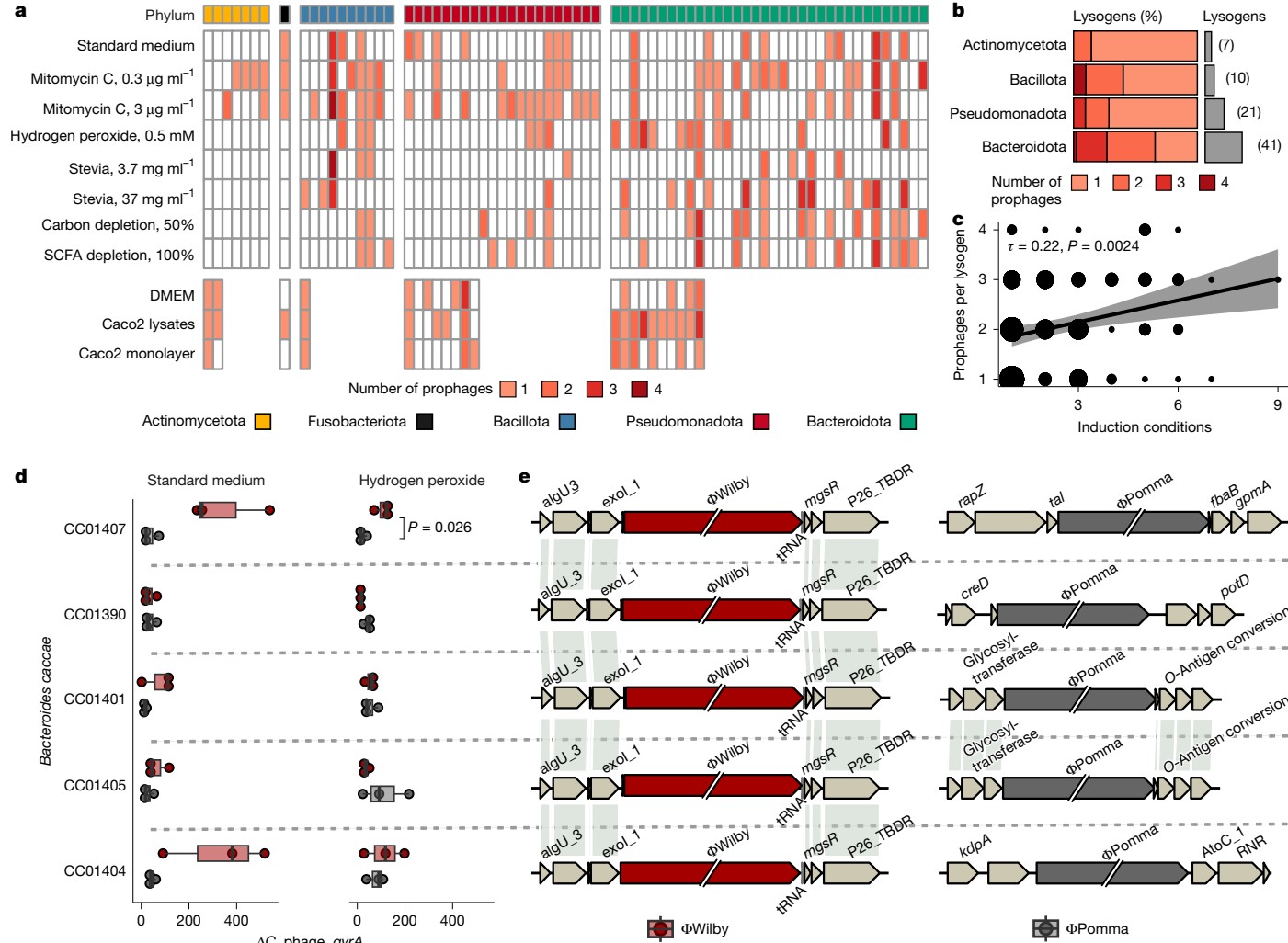

**Fig. 4 | Comparison of induction agents and analysis of polylysogeny within gut isolates. a**, The number of induced prophages per sample (condition in rows and isolates in columns). The isolate phylum is shown along the top bar. Actinomycetota is shown in yellow, Fusobacteriota in black, Bacillota in blue, Pseudomonadota in red and Bacteroidota in teal. **b**, The distribution of induced single and polylysogens per bacterial phyla. Fusobacteriota were excluded due to a single isolate. **c**, Kendall's rank correlation between number of inducible prophages within lysogens and the number of conditions in which each phage was detected as induced (the size is based on the number of observations). The black line is the best-fit line and the grey areas show the 95% confidence interval of linear regression. **d**, The fold change in induced prophages over background in isolates grown in standard medium (left, $n = 3$ biological replicates) or with the addition of hydrogen peroxide (right, $n = 3$ biological replicates). The box plots show the median (centre line), first and third quartiles (box limits), and the whiskers extend to ±1.5× the interquartile range; the means of three technical replicates are shown as dots. Normality was tested using the Shapiro–Wilk test, and significant $P$ values (shown above brackets) of normal data were calculated using two-sided paired $t$-tests. **e**, The genome location of prophage ΦWilby (red) and ΦPomma (grey). Shaded lines connect genes with 100% amino acid identity (AAI).

was triggered by human cell factors or was the result of spontaneous induction mirroring the shift in host taxa was unclear[36,40]. We therefore investigated human host factors in the absence of community effects, using 32 pure culture isolates exposed to Caco2 cell monolayers, Caco2 cellular lysates and DMEM cell culture medium alone. We observed a modest increase in induction with lysed Caco2 cellular products compared with cell culture medium or intact cells, suggesting that human cellular lysis products act as prophage induction triggers. This is in accordance with previous observations of temperate virion expansion found in patients with inflammatory bowel disease that are associated with increased inflammation and cell death[4].

Recent advances have highlighted the complexities governing prophage induction within natural environments, ranging from SOS-independent induction triggers, interprophage competition and interference by defence mechanisms[41–48]. Across our pure culture and community inductions, only a minority of predicted prophages was detected to undergo lytic replication. We therefore propose that,

although the genetic pool of integrated prophages within the gut is large, only a fraction of these will readily undergo lytic replication. This is consistent with previous studies estimating low induction rates within the human gut and reduced lytic infection rates of temperate gut phages[25,49]. Furthermore, we detect distinct gene enrichment patterns where non-induced prophage predictions encoded fewer structural and lysis-associated genes, indicating that a portion of high-quality predictions might consist of prophage remnants or poor prophage predictions. Moreover, non-induced predictions with high sequence similarity to experimentally induced prophages exhibited increased non-synonymous substitution rates in integrase and excision-related genes. Deletion of one of these genes in an active prophage led to complete abolishment of induction, providing evidence for a genetic pathway towards prophage domestication.

A considerable portion of our isolates (52%) with inducible prophages were polylysogens, harbouring more than one replicating prophage[48]. We found a positive correlation between polylysogeny and

successful prophage induction conditions, which is consistent with previous reports of phage anti-repressor proteins targeting non-cognate prophages leading to synchronized prophage induction[50]. Finally, we show that induction of polylysogenic prophages varied between near identical isolates, which correlated with divergent prophage integration sites within the host genome. Thus, prophage induction is complex and influenced by growth condition, polylysogeny and prophage integration site. In summary, we demonstrate the feasibility of culture-based approaches to provide insights into temperate phage biology and their interactions within human-associated commensals, and provide a validated collection of phage–host pairs for future use in synthetic biology, microbiome and biotechnological advances.

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

## Methods

### Bacterial culture conditions

A culture collection of 252 bacterial isolates previously isolated and sequenced from human gut samples was used for prophage induction[51]. All bacterial culture work was performed in yeast-extract casitone fatty acid (YCFA) medium at 37 °C under anaerobic conditions (Whitely A95 anaerobic workstation) containing 10% carbon dioxide, 10% hydrogen and 80% nitrogen[52]. Each isolate was streaked onto YCFA agar plates and incubated for 24 h before a single colony was inoculated in 1 ml YCFA medium in a 96-well plate and incubated for 24 h. Frozen stocks of the 96-well master plates were maintained in glycerol suspension (25%, v/v) at −80 °C. Before each induction, 96-well plates containing 1 ml YCFA were inoculated from the frozen master plate and grown overnight.

### Bacterial phylogeny and prophage prediction

A set of 40 single-copy marker genes were extracted from the 252 bacterial isolates using progenome-classifier[53] and translated into amino acid sequences using SeqKit[54] (v.2). The protein sequences were concatenated and aligned using MAFFT[55] (v.7.310) before gaps were trimmed with trimAI[56] (v.1.4.1). Maximum-likelihood trees were constructed using RAxML[57] (v.8.2.12) PROTGAMMALGF model with 100 bootstraps replicates and visualized in iTOL[58]. Bacterial clusters sharing 99% ANI were identified using dRep[59] (v.3.0.0) with the '-pa 0.9 --sa 0.99' flags. Prophage regions were predicted using Virsorter[60], Vibrant[61] (default settings) and VirFinder[62] (minimum length 5 kb; 0.7 score; and P value 0.05). Completeness was predicted using CheckV[63] and contaminating bacterial regions were removed. Trimmed predictions were located within their cognate bacterial genome and overlapping predictions were merged using R IRanges[64] (v.2.28.0).

### Pure culture prophage induction and sequencing

Prophages were induced by one of two methods. (1) Overnight starter cultures were diluted 1:50 in 1.5 ml standard YCFA medium and grown for 5 h before the addition of mitomycin C[19] (0.3 or 3 µg ml⁻¹, M4287, Sigma-Aldrich) or hydrogen peroxide[20] (0.5 mM, H1009, Sigma-Aldrich). (2) Starter cultures were diluted 1:50 directly into standard YCFA medium, YCFA medium supplemented with Stevia[23] (3.7 or 37 mg ml⁻¹, SweetLeaf, organic Stevia leaf extract), carbon-depleted medium[22] (YCFA medium with 50% reduced carbon source) or SCFA-depleted[21] medium (YCFA medium without SCFAs). All cultures were then grown for 20–25 h followed by centrifugation at 4,000$g$ for 30 min and 1 ml supernatants were collected. The supernatants were treated with 10 µg ml⁻¹ DNase I (DN25, Sigma-Aldrich) and 0.01 volume RNase A (R6148, Sigma-Aldrich) for 1 h at 37 °C. Viral particles were precipitated in 7% PEG 8000 0.3 M NaCl overnight at 4 °C, followed by centrifugation at 14,000$g$ for 30 min, after which the pellets were dissolved in 50 µl TE buffer at 4 °C. Next, 20 µl of each sample was mixed with 5 µl loading dye containing 0.8% SDS and 60 mM EDTA and heated at 65 °C for 10 min. The samples were loaded onto 0.4% agarose gels and run in TAE for 1.5 h, followed by visualization of phage sized (about 50 kb) DNA bands using Image Studio Lite (LI-COR Biosciences) with a sample to control well signal ratio cut off of 0.03 (refs. 65). Samples with suspected viral DNA were treated with 0.5% SDS and 100 µg ml⁻¹ proteinase K at 55 °C for 1 h followed by a 10 min inactivation at 65 °C. Phenol–chloroform–isoamyl alcohol (25:24:1) extraction was performed followed by sodium acetate (0.3 M final) and 70% ethanol precipitation with 0.4 mg ml⁻¹ glycogen overnight at 4 °C. The DNA quantity was validated using Qubit (Thermo Fisher Scientific), with a minimum of 2 ng µl⁻¹ required for sequencing. From these, a subset of samples was selected for sequencing as follows. First, all of the samples grown in standard YCFA medium or induced with mitomycin C were selected. Second, samples from at least one isolate within a bacterial cluster (99% ANI) grown in the remaining five induction conditions were selected, except for the Fusobacteria isolate, which was sequenced only in standard medium and mitomycin

C. For 17 out of 84 clusters, more than one isolate was sequenced in all conditions. Nextera-XT libraries were constructed and sequenced on either the Illumina NextSeq 2000 or Illumina NextSeq 550 system.

### Human cell culture prophage induction

Human colonic epithelial immortalized cells (Caco2 TC7[66]; genotype verified by the AGRF Human Cell Line Identification Service) were maintained in Dulbecco's modified Eagle medium (DMEM, low glucose, GlutaMAX supplement, pyruvate) (Thermo Fisher Scientific), containing 10% FCS (Bovogen) in 5% $CO_2$ at 37 °C in T75 cm² flasks (Nunc, Thermo Fisher Scientific). Cells were routinely tested for the presence of mycoplasma contamination (MycoStrip, Invivogen) and were confirmed to be mycoplasma negative. For sonicated cell inductions, $4 \times 10^7$ cells in 5 ml DMEM (without FCS) were sonicated on ice (5 cycles, 30 s on and off at 40% frequency). Sonicated Caco2 cells were confirmed by observing efficient lysis of cells by bright-field microscopy and stored at −20 °C. Before the addition of bacterial cells, sonicated cells were thawed on ice and pre-reduced DMEM was resuspended with the cells for a final cell density of $5 \times 10^5$ cells per ml in the Whitley A95 Anaerobic Workstation (Don Whitley Scientific) at 37 °C. Sonicated cells in DMEM (2 ml) were added to 6-well tissue culture plates to achieve a total of $1 \times 10^6$ cells per well.

Two days before induction, $1 \times 10^6$ Caco2 cells were seeded in 6-well tissue culture plates (Nunc, Thermo Fisher Scientific) in a final volume of 2 ml per well with DMEM containing 10% FCS and incubated for 48 h at 37 °C under 5% $CO_2$. Caco2 confluent cell layers were serum starved under anaerobic conditions for 2 h by replacing the cell medium with 2 ml pre-reduced DMEM (no FCS) per well in the Whitley A95 anaerobic workstation at 37 °C.

Individual working stocks of bacterial isolates were prepared from overnight bacterial cultures that were centrifuged at 4,000$g$ for 10 min, resuspended to an optical density at 600 nm ($OD_{600}$) of 5.5, combined 1:1 with 150 µl 50% glycerol and stored as frozen glycerol stocks (25% final glycerol). Glycerol stocks of individual bacterial isolates were thawed and 8.5 µl of each isolate was added to 6-well tissue culture plates containing 2 ml pre-reduced DMEM medium only, 2 ml sonicated Caco2 cells (total cell number, $1 \times 10^6$ cells per well) or confluent Caco2 cell layer. All cultures were grown for 24 h followed by centrifugation at 4,000$g$ for 30 min after which 1.8 ml supernatants were collected, and viral DNA was extracted and sequenced on the Illumina NextSeq 2000 system as previously described.

### Regions of interest

Reads were trimmed using Trimmomatic[67] (v.0.38) (SLIDINGWINDOW:4:25 MINLEN:100) and used to identify induced prophages using two approaches. First, high-quality prophage predictions (>50% completeness) were validated for induction as follows. Read coverage for each library were obtained on their corresponding genome using Bowtie2 (ref. 68) (v.2.3.5) (default settings). Genome coverage in 100 bp increments was obtained using Samtools[69] (v.1.9) and Deeptools[70] (v.3.1.3) and the average modified $z$ score, coverage fold increase and Cohen's $D$ of prophage regions was calculated as follows:

$$z\text{-score}_{ave} = \text{mean}(0.6745 \times (x_p - \tilde{x})/\text{median}\lceil x_h - \tilde{x}\rceil) \qquad (1)$$

where $z$-score$_{ave}$ is the average $z$ score of the predicted region, $x_p$ is 100 bp coverage increments of the phage region, $x_h$ is 100 bp coverage increments of the host and $\tilde{x}$ is median coverage of the host and

$$\text{Cohen's } D = \frac{\text{mean}(x_h) - \text{mean}(x_p)}{\sqrt{\frac{S_h^2 + S_p^2}{2}}} \qquad (2)$$

where Cohen's $D$ is the effect size of prophage versus host coverage, $S_h$ and $S_p$ is the s.d. of the host and phage coverage, respectively[71]. Regions

with a minimum average modified $z$ score of 3.5 or an average twofold coverage and Cohen's $D$ larger than 0.7 were retained. A custom Python script was then used to refine the start stop positions of the prophage regions within each genome, removing flanking 100 bp increments with coverage less than 25% of the mean prophage coverage (code is available at https://doi.org/10.26180/29946902.v1). In a second approach, regions of increased coverage were identified without previous prophage predictions using hafeZ[72] (v.1.0.2) (default settings with the -N -S flags). Some of the identified regions of interest were found to be split across several host contigs. To resolve these into full-length phage contigs, de novo assembly using MetaViral SPade[73] (default setting) was performed and contigs overlapping with hafeZ prediction were retained. The resulting contigs were dereplicated at 99% ANI over 85% of the AF using scripts from the CheckV repository and the longest representative of each prophage were retained for further analysis.

### Identification of induced temperate phages

Proteins from the resulting contigs were predicted and annotated using PROKKA[74] (v.1.14.6) (default settings, --hmms) with the PHROGS[38] database. Furthermore, all proteins were scanned against the hmm databases provided by Cenote-Taker2 (ref. 75) using Hmmer[76] (v.3.3.1) hmmscan (-E 1e-9). To remove potential fragmented protein hits against hallmark genes, a custom phage database was constructed of genomes from refs. 9,31 and the INphared database (December 2021 version) dereplicated at 95% ANI over 85% AF using CheckV scripts[77]. The same HMM searches were performed on proteins from the database and half the average length of the middle 80% percentile was calculated as a cut-off for each *Caudoviricetes* hallmark gene (265 amino acids for terminase large subunit, 245 amino acids for portal protein and 186 amino acids for major head protein). To identify any non-*Caudoviricetes* genomes, HMM searches for *Microviridae*, *Tectiviridae* and *Inoviridae* were performed as follows. *Microviridae* hallmark VP1 proteins from ref. 78 was made into HMM profiles using MAFFT v.7.310 with the standard settings followed by TABAJARA (-t 0.5 -p 50 -w 15 -b 15 -mb 15 -m c -gs 20 -md 3 -cs yes -mb 20)[79]. Multiple-sequence alignments of the double jelly-roll hallmark protein of *Tectiviridae* were obtained from a previous study[80] and turned into HMM profiles using HMMER v.3.3.1 hmmbuild. These and the *Inoviridae* morphogenesis protein family HMMs provided by ref. 81 were searched against all proteins using hmmscan (-E 1e-9 and score ≥ 30). Contigs containing at least one viral hallmark gene were retained.

### Synthetic microbiome community prophage induction

Working stocks of the synthetic microbiome community were prepared by combining 1.5 ml of overnight bacterial culture diluted to $OD_{600}$ 0.7, after which the community was centrifuged at 8,000$g$ for 10 min, resuspended in 10 ml fresh YCFA and stored as frozen glycerol stocks (25% final glycerol). Human Caco2 confluent cell layers grown for 48 h in 175 cm$^2$ tissue culture flasks (NUNC) in DMEM containing 10% FCS with 5% $CO_2$ at 37 °C were transferred to anaerobic conditions (Whitley A95 anaerobic workstation) and the medium in each flask was replaced with 70 ml pre-warmed and pre-reduced (overnight) DMEM (without FCS). Next, 200 μl of the frozen community stock was added to each 70 ml tissue culture flasks containing Caco2 cell layers as well as to culture bottles containing 200 ml pre-reduced YCFA medium, both in five replicates. All cultures were grown anaerobically with samples (14 ml) taken at 24, 48 and 72 h. Total metagenome DNA extraction was performed using the FastDNA SPIN Kit for Soil (MP Biomedicals) on 1 ml of each sample. The remainder of the sample was centrifuged at 3,000$g$ for 30 min and the supernatant was collected and filtered through 0.45 μM syringe filters (Acrodisc, Pall) followed by incubation for 15 min with 0.1 volumes of chloroform at room temperature. The samples were then centrifuged at 4,000$g$ for 15 min and 9 ml of the aqueous phase was collected and treated with 10 μg ml$^{-1}$ DNase I (DN25, Sigma-Aldrich) and 120 μl RNase A (R6148, Sigma-Aldrich) for

1 h at room temperature. Viral particles were precipitated using 7% PEG 8000 incubated overnight at 4 °C, centrifuged at 12,000$g$ for 1 h and resuspended in 100 μl TE buffer. Viral DNA was extracted and sequenced on the Illumina NextSeq 2000 system as previously described.

### Synthetic microbiome community prophage detection

Reads from the synthetic microbiome community were trimmed using Trimmomatic[67] (v.0.38) (SLIDINGWINDOW:4:25 MINLEN:100), decontaminated of human reads (GCA_000001405.29) using Bowtie2 (v.2.3.5) and Samtools (v.1.19) and de-interleaved using bbmap (v.39.06). A database of community prophage genomes (high-quality predictions $n$ = 338) and bacterial host genomes ($n$ = 78, masked for prophage regions using bedtools (v.2.26.0)) was constructed[70]. Decontaminated community reads from samples were aligned against the database using Bowtie2 (default flags) and coverage was obtained using Samtools 'coverage'. Host abundance was calculated from the Samtools outputs of total DNA extracted sequence libraries.

In viral-enriched samples, prophage species were regarded to be induced if at least one representative genome was covered with reads ≥85% of the length with a twofold increase in coverage (depth) over the mean host genome coverage (sum coverage of host contigs normalized by length) in a minimum three out of five replicates. For detection of individual prophage genomes, a custom metagenomic read classification database was built using KrakenUniq (v.1.0.4)[82] containing prophage and prophage-masked host genomes. For database construction purposes, phage sequences were assigned the NCBI taxonomy IDs of their host bacteria. NCBI taxonomic data were downloaded using the 'krakenuniq-download --db taxonomy' command on 7 July 2024. The KrakenUniq database was constructed with the krakenuniq-build command using Jellyfish (v.1.1.12) for $k$-mer extraction, a $k$-mer size of 21 and the --taxids-for-genomes option. Paired-end reads were merged using the read_merger.pl script within KrakenUniq and subsequently classified using the krakenuniq command with default parameters. On the basis of data from pure isolate inductions, a cut-off of 0.25 $k$-mer coverage, 10 reads and 100 unique $k$-mers was selected for calling detection of phage and a cut-off of 10 reads and 18,000 unique $k$-mers for calling detection of bacterial host genome was used. Prophage genomes were regarded as induced if they had a twofold increase in kmerDuplicity over mean host genome kmerDuplicity (sum duplicity of host contigs normalized to length) in a minimum of three out of five replicates. Prophages from undetected isolates were regarded as induced if detected in a minimum of three replicates. To estimate the number of detectable prophages within the community (prophages with at least 100 unique $k$-mers), pairwise distances between all prophage and prophage-masked host genomes were calculated using Mash (v.2.2.2) with a $k$-mer size 21 and sketch size 5000 (ref. 83). Neighbour-joining was applied to the resulting distance matrix as implemented in RapidNJ (v.2.3.2) with default parameters[84]. Using the resulting tree, a metagenomic classification database was constructed using Expam (v.1.2.2.5) with a $k$-mer size 21 and number of unique $k$-mers per genome was obtained using the CountUniqueKmers.py script (https://github.com/seansolari/expam/scripts/database/CountUniqueKmers.py)[85].

### Taxonomic annotation and DGR identification

Viral taxonomy was assigned based on a combination of the protein alignment method previously described[2] against the INphared database and genus level clustering using vContact2[29] against phage genomes in the custom made database used for hallmark gene searches. In cases in which the taxonomic assignments from the protein voting and genus-level clustering method differed, the lowest common classification was assigned. Species level dereplication was performed at 95% ANI over 85% AF using scripts from the CheckV repository. DGRs were identified using DGRscan[86] with the default settings, and remote VRs were identified querying the template repeat using BLASTn (v.2.7.1+) (-dust no -perc_identity 75 -qcov_hsp_perc 50 -ungapped -word_size 4).

DGR-positive genomes were defined as genomes encoding both a reverse transcriptase gene and containing repeat regions.

## Metagenomic read mapping

The fractional abundance and prevalence of induced prophages within gut viromes were performed as described previously[9] using the 1,241 human gut viromes described therein. Reads from each virome were competitively aligned to the temperate phage species genomes together with the custom database previously described. The number of reads and read coverage was obtained using Samtools 'coverage' (v.1.9). The fractional abundance of a genomes was calculated as follows:

$$\text{Fractional abundance} = \frac{\text{reads}_{\text{genome}}/\text{length}_{\text{genome}}}{\text{Total reads}_{\text{virome}}/50{,}000} \qquad (3)$$

and the sum fractional abundance was normalized to 1 as previously described[87]. A genome was counted as present within a virome if at least 70% of the genome length was covered by reads.

## Analysis of non-induced prophages

Proteins of predicted prophages were predicted using PROKKA v.1.14.6 (default setting, --hmms) and annotated using the PHROG database. The total gene counts per PHROG category and the presence–absence of PHROG categories within each genome were obtained for induced, high-quality predictions (>50% completeness) and low-quality predictions (<50% completeness). The percentage gene frequency change of PHROG categories between induced, high-quality and low-quality predictions was calculated for total genes and presence–absence counts as follows:

$$\text{Frequency change}(\%) = \frac{100 \times (f_{\text{cry}} - f_{\text{in}})}{f_{\text{in}}} \qquad (4)$$

where $f_{\text{cry}}$ and $f_{\text{in}}$ are the gene frequencies in the high completeness prediction and induced prophage set, respectively.

High-quality predictions were aligned to induced prophages using BLASTn (v.2.15.0+) and pairs with a minimum of 95% ANI over 85% AF (Checkv anicalc.py[63] script) were further filtered to only include hosts that had been sequenced in the same condition(s) as the induced prophage. The same search was performed to identify induced–induced prophage pairs. The number of HGT and insertion–deletion events between the pairs was calculated using R IRanges (v.2.28.0) and splicejam (v. 0.0.77) packages, where an HGT event was defined as a gap of a at least 50 bp within the alignment present in both pairs and insertion–deletion events was defined as gap (minimum 50 bp) present in one of the pairs but not the other[88]. Gaps involving the ends of either prophage were excluded. Host ANI of prophage pairs was calculated using fastANI[89] (v.1.33). dN/dS ratios between prophage pairs was calculated using dRep (compare --SkipMash --S_algorithm goANI) and the dnds_from_drep.py[90] script.

## Inactivation of ΦPomma in *B. faecis* isolate CC01414

Gene deletion of the DNA transposition protein gene within ΦPomma in *B. faecis* isolate CC01414 was achieved using the CRISPR–Cas-based system described previously[91]. First, we redesigned pB025, which contains the FnCas12a system, with a sgRNA targeting the DNA transposition protein gene (gene location, base pairs 5610–6524) along with a repair template containing 1,000 bp of homologous DNA up and downstream of this gene (Supplemental Table 10). This plasmid (pB025_09) was transformed into competent *E. coli* S17 and grown aerobically in LB medium supplemented with 100 μg ml⁻¹ ampicillin. *B. faecis* was grown in brain–heart infusion (BHI) liquid medium supplemented with haemin, resazurin and vitamin K3 (menadione) under anaerobic conditions. Conjugation was performed under anaerobic conditions and

*B. faecis* transconjugants were selected for with 200 μg ml⁻¹ gentamicin and 25 μg ml⁻¹ erythromycin. A deletion mutant was identified before anhydrotetracycline (aTc) induction, presumably due to leaky expression. This deletion mutant was verified by PCR and sanger sequencing confirming the clean deletion of the DNA transposition protein gene (CC01414 Δ*tran*).

Induction of ΦPomma in CC01414 wild type and deletion mutant (Δ*tran*) was performed as described previously using hydrogen peroxide (0.5 mM) and stevia (37 mg ml⁻¹) using three separate induction reactions for each condition and isolate. Lysates were treated with 2% chloroform, centrifuged for 20 min at 4,000*g* at 4 °C, DNase treated and phage precipitation with PEG, DNA extraction and sequencing was performed as described previously. qPCR of DNA extracted phage lysates was performed in technical triplicates using SYBR Green I Master Mix (Roche Diagnostics) with the Roche Lightcycler 480 system containing 1 μM of each primer, 2 μl of DNA template and 1× SYBR Green I Master Kit, in a final reaction volume of 20 μl. Cycle parameters were as follows: initial denaturation at 95 °C for 10 min; followed by 45 cycles of 95 °C for 20 s, 62 °C for 20 s, and 72 °C for 30 s. Primers were designed using Primer blast and no cross reactivity to bacterial background was detected (https://www.ncbi.nlm.nih.gov/tools/primer-blast). The standard curve was produced with gBlock sequence from IDT containing the sequence targeted by ΦPomma primers.

## Differential prophage induction qPCR

Isolates were streaked onto YCFA plates and grown for 24 h. Three sperate colonies from each isolate were inoculated into 1 ml YCFA broth and grown overnight. Overnight cultures were diluted 1:50 into 1.5 ml standard YCFA medium and hydrogen peroxide was added after 5 h of growth. All cultures were grown for an additional 20 h and lysates were treated with 2% chloroform and centrifuged for 20 min at 4,000*g* at 4 °C and frozen at −80 °C until analysis was performed. qPCR was performed as previously described using 5 μl of DNA template and annealing temperature of 60 °C for 30 s and elongation of 30 s. qPCR primer pairs were custom designed using Primer 3 (https://primer3.org/). In silico PCR amplification (http://insilico.ehu.eus/user_seqs/PCR/) did not show cross reactivity of primers to the rest of the bacterial genome. Standard curves for primer efficiency analysis were generated by tenfold dilution in PCR-grade $H_2O$. Samples were diluted tenfold and qPCRs was performed in triplicates. The efficiency of each primer calculated as in equation (5) and corrected $\Delta C_t$ values calculated as in equation (6):

$$\text{Efficency} = 10^{-1/\text{slope}} \qquad (5)$$

$$C_t = \frac{\text{Efficiencyt}_x^{C_{t,x}}}{\text{Efficiencyt}_y^{C_{t,y}}} \qquad (6)$$

## Long-read sequencing

Isolates were streaked onto YCFA plates and grown for 24 h. Single colonies were grown overnight in 40 ml of YCFA medium, pelleted by centrifugation at 4,000*g* for 10 min and washed four times in 1 ml of PBS. DNA was extracted using the Monarch HMW DNA extraction kit (New England Biolabs) according to the Gram-positive bacteria protocol, with modifications. Cells were lysed in 300 μl of STET buffer (8% sucrose 5% Triton X-100 50 mM EDTA, 50 mM Tris pH 8) containing 10 mg ml⁻¹ lysozyme, 300 μl of HMW gDNA tissue lysis buffer and 20 μl of proteinase K and incubated at 56 °C for 10 min. The lysates were treated with 10 μl of RNase A at 56 °C for 5 min followed by 300 μl of protein separation solution. The samples were mixed by inversion for 2 min then centrifuged at 4 °C for 20 min at 16,000*g*. The supernatants were collected and 550 μl of isopropanol was added to 800 μl of supernatant. The samples were inverted for 5 min, or until DNA was

precipitated, and DNA was pelleted by centrifugation at 4 °C for 10 min at 12,000*g*. The resulting pellet was washed twice with 500 µl of gDNA wash buffer and resuspended in nuclease free water. Library preparation and Oxford Nanopore MinION sequencing was performed using either the Oxford Nanopore ligation sequencing kit (SQK-LSK109) with native barcoding expansion kit (EXP-NBD114) (CC01407, CC01390, CC01401 and CC01405) or the rapid barcoding kit 96 (SQK-RBK110.96, CC01404). Resulting long reads were hybrid assembled with Illumina short reads into closed genomes using dragonfly (v.1.0.14) (CC01407, CC01390, CC01401 and CC01405) or into near complete genome using unicycler[92] (v.0.4.7) (CC01404) with subsequent scaffolding using RagTag[93] with CC01407 genome as reference.

### Statistical analysis and visualization
Significance of PHROG gene category was calculated with Fisher's exact test (two sided) and *P* values were adjusted with the Hochberg method using R base stats (R v.4.1.3) and rstatix (v.0.7.0) packages. Pearson's correlation test between host ANI and phage pair inducibility as well as Kendall's rank correlation between the number prophages within lysogens and prophage inducibility was calculated and plotted using the R ggpubr (v.0.4.0) package. Significance of horizontal gene transfers and dN/dS data was calculated with Wilcoxon rank-sum test (two sided) and adjusted by Hochberg method using R ggpubr (v.0.4.0) and rstatix (v.0.7.0) packages. Normality of qPCR fold change between induced prophage in polylysogens calculated with Shapiro–Wilk test and significance tested with paired *t*-test (two sided) using rstatix (v.0.7.0), preferential induction calculated with Wilcoxon rank-sum test (two sided) and variance of means between isolates was calculated with ANOVA using the R base stats (R v.4.1.3) package. Genome maps were visualized using the R gggenomes (v.0.9.9.9000) package, and genome read coverage was visualized using R ggplot2 (v.3.5.1).

### Reporting summary
Further information on research design is available in the Nature Portfolio Reporting Summary linked to this article.

## Data availability
All data for this study have been deposited in the European Nucleotide Archive (ENA) at EMBL-EBI under project number PRJEB64565 and accession numbers are provided in Supplementary Table 1, 3 and 12. Bacterial isolates are available through the Australian Microbiome Culture Collection (AusMiCC). Metadata of viromes used in this study along with accession numbers are provided in Supplementary Table 5. Bioinformatic scripts and figure data are available at Figshare[94] (https://doi.org/10.26180/29946902.v1). Illustrations were made using Inkscape (https://inkscape.org).

## Code availability
This study did not generate any new code. Analysis scripts are available at Figshare[94] (https://doi.org/10.26180/29946902.v1).

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

**Acknowledgements** This work was supported by the Australian Research Council (ARC) Discovery Project grant (DP210103296). S.D. and L.A.-F. were supported by Monash University Postgraduate Research Scholarship funding their doctoral studies; R.A.E. by an award from the National Institute of Health (NIH NIDDK; RC2DK116713) and awards from the ARC (DP220102915, DP250103825, and FL250100019); S.C.F. by a CSL Centenary Fellowship; and J.J.B by a National Health and Medical Research Council (NHMRC) Investigator Grant Leadership Level 1 (NHMRC; 2026130) and Monash University Research Talent Accelerator (RTA) 2023 program. We acknowledge the staff at the Hudson Genomics Facility, Monash eResearch Team and the Victorian State Government Operational Infrastructure Scheme for support.

**Author contributions** J.J.B. and S.C.F. conceived and designed the study. R.A.E., D.L. and J. A. Grasis contributed ideas and expertise. S.D. performed in vitro inductions, molecular work, sequencing, informatic analyses and most of the data preparation. L.A. assisted with informatic analyses, data interpretation, molecular work and sequencing. C.K. performed DNA extractions. D.S. performed qPCR assays. R.B.Y., J. A. Gould and E.L.R. assisted with molecular work and sequencing. E.L.R., E.L.G. and R.B.Y. isolated and assisted with the cultured bacterial isolates. N.C. assisted in Bacteroides gene editing. C.J.R.T. and N.N.-A. assisted with identification of induced prophages. E.L.R., E.L.G., J.J.B., C.A.H.D. and M.C. performed synthetic microbiome co-culture and pure-culture inductions. S.S. assisted in design of analysis methods for synthetic microbiome inductions. S.D., S.C.F. and J.J.B. wrote the paper. J.J.B. supervised all aspects of the work and all of the authors approved the final manuscript.

**Funding** Open access funding provided by Monash University.

**Competing interests** S.C.F., S.S. and R.B.Y. are advisors to or employees of Biomebank. The other authors declare no competing interests.

## Additional information
**Correspondence and requests for materials** should be addressed to Jeremy J. Barr.

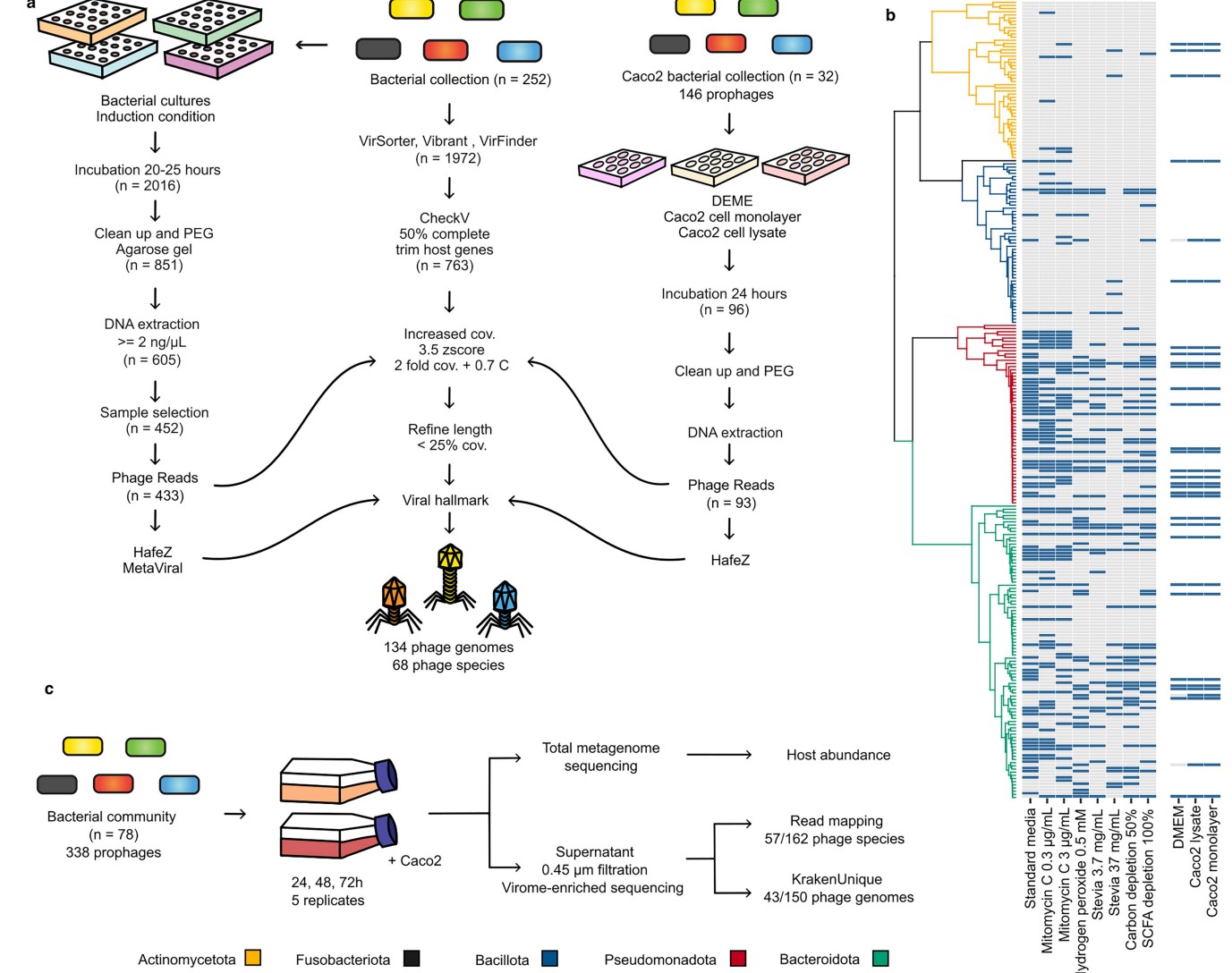

**Extended Data Fig. 1 | Methods overview single isolate and community induction. a**, Schematic of bacterial single isolate inductions. Isolates were grown 20–25 h in standard media, Mitomycin C (0.3 and 3 μg/mL), hydrogen peroxide (0.5 mM), Stevia (3.7 and 37 μg/mL), carbon depleted or SCFA depleted media (n = 2,016). Samples were centrifugated, DNAse treated, PEG concentrated, and phage sized DNA bands were detected on agarose gels. DNA was extracted from samples with phage sized bands (n = 859) and DNA concentrations >2 ng/μL were chosen for sequencing based on condition and bacterial cluster (n = 452), 19 of which failed library preparation. Second, a subset of isolates (n = 32) were induced in DMEM media with or without sonicated or intact human colonic epithelial cells (Caco2) and DNA was extracted (n = 96), 3 of which failed library preparation. Prophages were predicted using Virsorter, Vibrant and VirFinder (n = 1972) and predictions >50% complete were trimmed for bacterial flanking regions (n = 736). Prophage reads (n = 526) were aligned to host genome and predictions with >2-fold coverage and Cohen's D > 0.7 or mean zscore >3.5 were retained. Further, reads were investigated for induction using HafeZ. De novo assembly using MetaViralSpades was used to resolve predictions spanning host contigs. Predictions were dereplicated at 99% ANI and 85% AF retaining the longest representative with at least one viral hallmark gene. **b**, Phylogenetic tree of bacterial isolates with heatmap depicting sequenced phage-enriched samples (shown in blue); Actinomycetota yellow (20/409), Fusobacteriota black (6/8), Bacillota blue (37/414), Pseudomonadota red (216/498) and Bacteroidota teal (247/786). **c**, Schematic of the synthetic community inductions. Isolates (n = 78) were co-cultured with or without Caco2 cell monolayer for 24, 48 and 72 h. Host abundance was calculated on total metagenome DNA using read mapping. Prophage induction was detected in phage enriched DNA at species level using read mapping as well as on individual prophage using KrakenUnique.

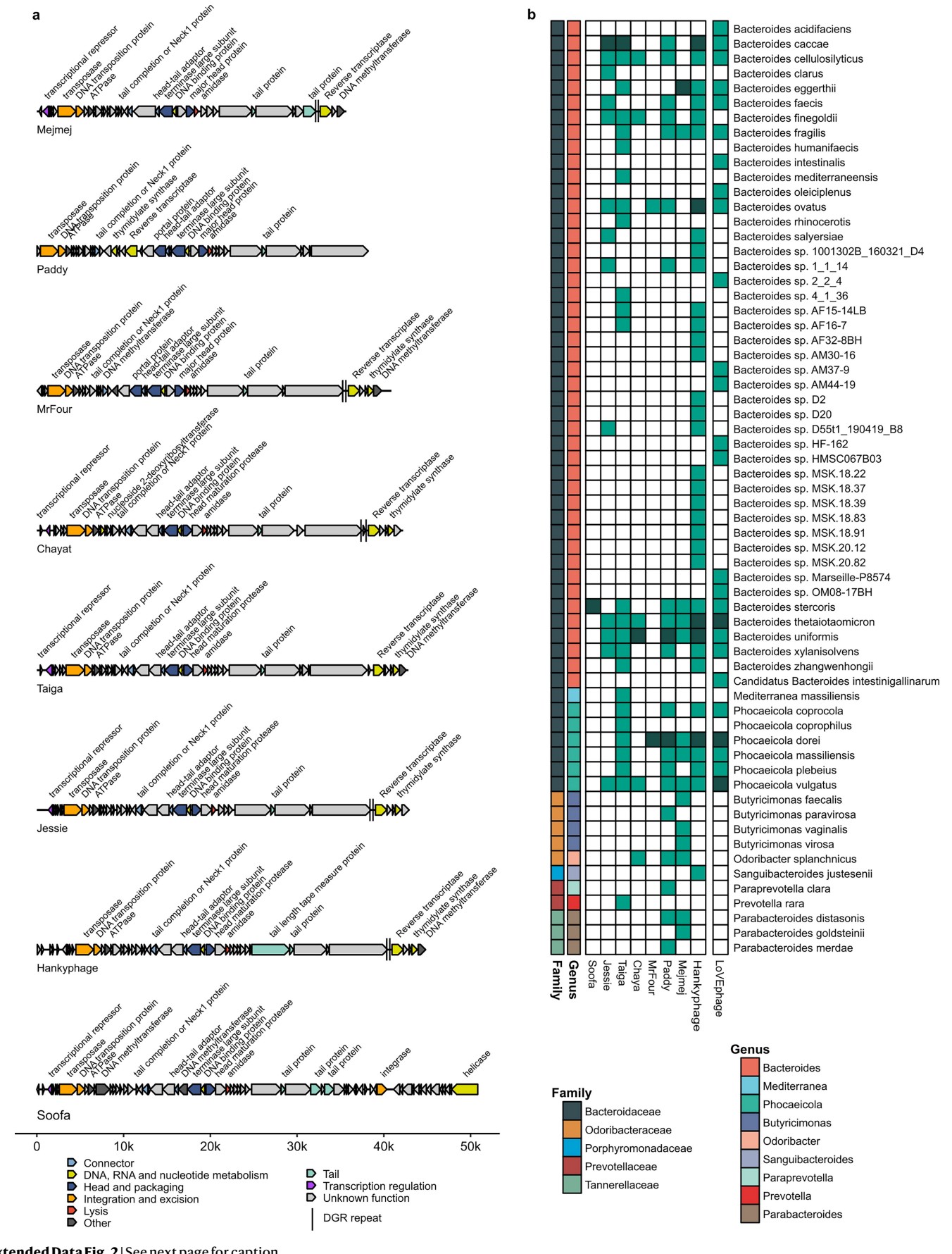

**Extended Data Fig. 2** | See next page for caption.

**Extended Data Fig. 2 | Hankyvirus genus overview and host range of Hankyvirus and LoVEphage. a**, Annotated genome maps of the eight Hankyvirus genus phages species. Genomes are scaled by length with the ruler displayed at the bottom of the figure. Genes coloured by PHROG categories and unknown genes coloured in grey. DGR template and variable repeats denoted with black lines. **b**, Host range of Hankyvirus and LoVEphage species induced in this study. Light green denotes hosts found in NCBI RefSeq bacterial genome database; dark green denotes hosts in which the phage was actively induced in this study.

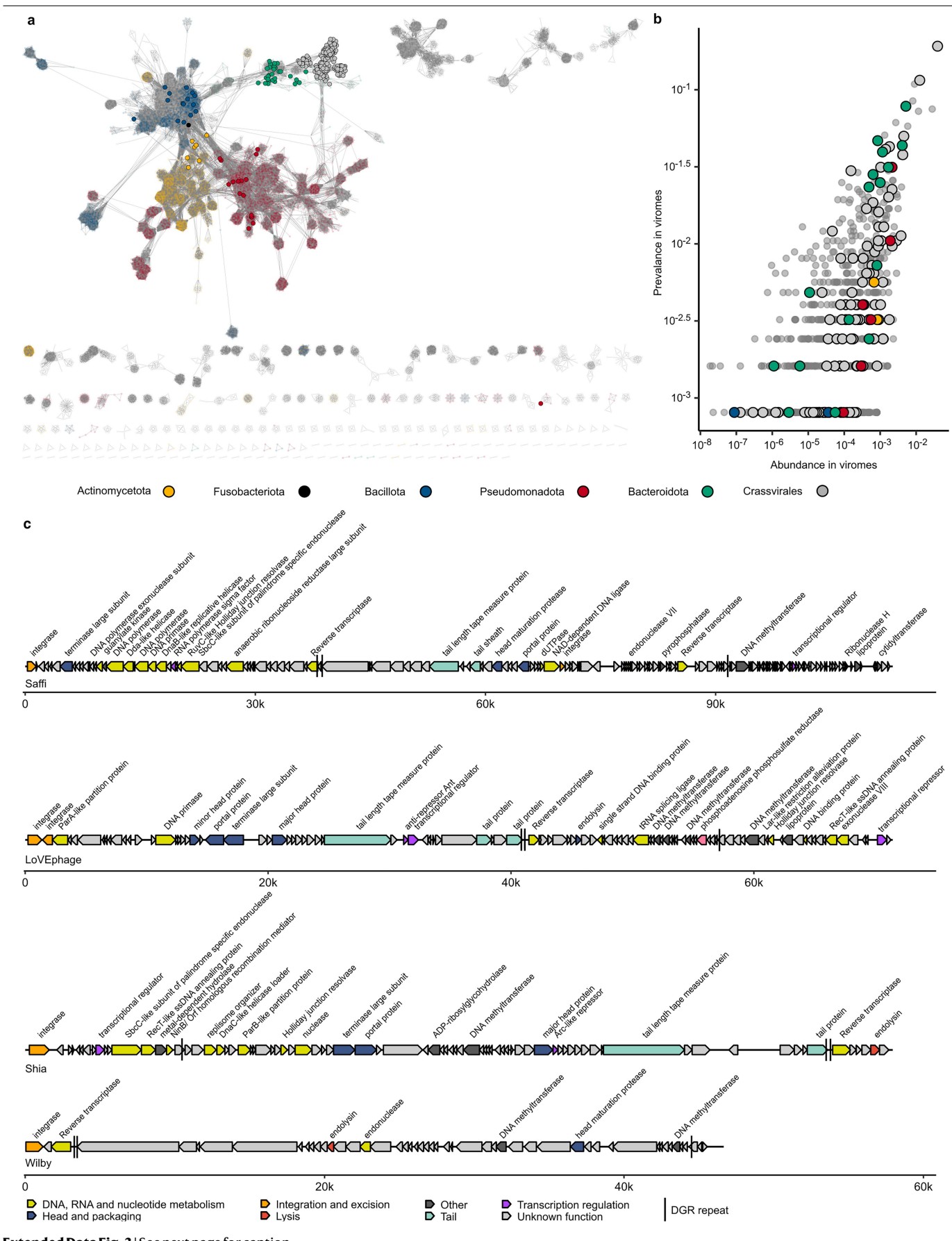

**Extended Data Fig. 3** | See next page for caption.

**Extended Data Fig. 3 | Phage species diversity and genomes with double variable repeat DGRs. a**, Gene sharing network between induced prophage species (solid circles, n = 68) and database representatives (9920). Representatives in translucence and coloured by host phyla when applicable otherwise in grey, except *Crassvirales* genomes from Yutin et al. 2021 which are highlighted with solid black boarded grey circles[31]. **b**, Mean fractional abundance and prevalence of Caudoviricetes phages within gut viromes (1241) shown as in Fig. 2b, but highlighting *Crassvirales* genomes in large solid grey circles. **c**, Annotated genome maps of four phage species encoding a second variable repeat (VR) distal from the RT cassette. Genes coloured by PHROG categories and unknown genes in grey. DGR template and variable repeats denoted with black lines.

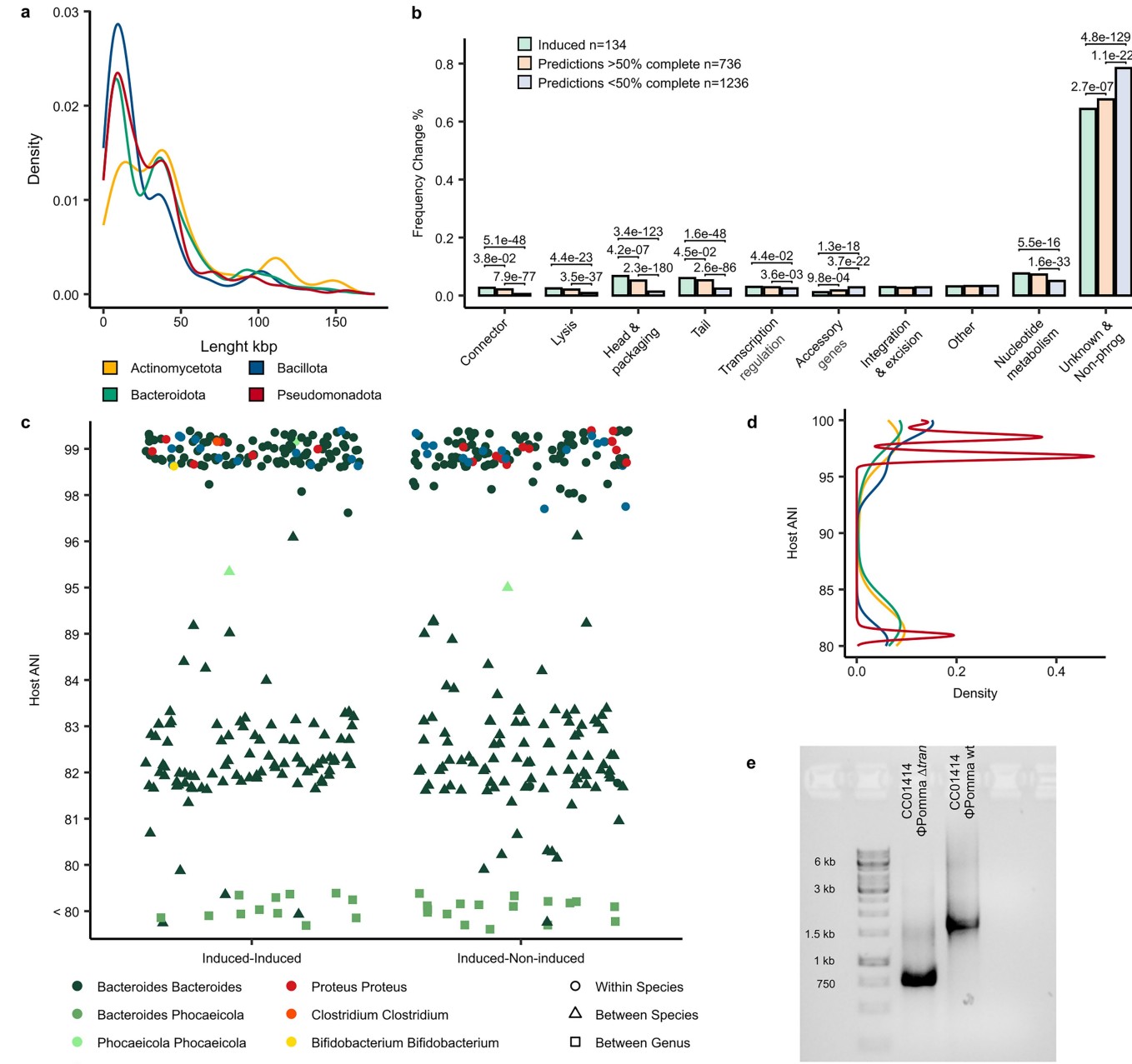

**Extended Data Fig. 4 | Comparison of induced versus predicted prophage sequences and host nucleotide identity. a**, Length distribution of all predicted prophages in this study separated by host phyla. Fusobacteriota excluded as only single isolate. A bimodal length distribution was observed for all phyla, with an initial peak at around 8 kb followed by a second peak at around 37 kb. **b**, Frequency of PHROG gene categories across induced (green, n = 134), high quality (> 50% completeness, orange, n = 736) and low quality (< 50% completeness, blue, n = 1236) predictions. Significant *p* values calculated using Fisher's exact test (two sided) and adjusted by Hochberg method shown above bars. **c**, Host ANI comparisons of induced to induced (n = 222) and induced to non-induced (n = 231) isolates with high similarity prophage pairs. Comparisons coloured by genus and shape based on NCBI taxon. Host pairs with less than 80% similarity grouped separately as too divergent for reliable ANI score. **d**, Density of ANI comparisons of all isolates within the dataset separated by phyla, coloured as in panel a. **e**, PCR amplification with phage specific primers surrounding the targeted gene (DNA transposition protein), visualized by agarose gel electrophoresis (single experiment) alongside GeneRuler 1 kb DNA ladder, showing deletion of a ~ 900 bp fragment in øPomma Δ*tran* mutant (gene product size, 915 bp).

# Reporting Summary

## Statistics

For all statistical analyses, confirm that the following items are present in the figure legend, table legend, main text, or Methods section.

| n/a | Confirmed | |
|---|---|---|
| ☐ | ☒ | The exact sample size (*n*) for each experimental group/condition, given as a discrete number and unit of measurement |
| ☐ | ☒ | A statement on whether measurements were taken from distinct samples or whether the same sample was measured repeatedly |
| ☐ | ☒ | The statistical test(s) used AND whether they are one- or two-sided *Only common tests should be described solely by name; describe more complex techniques in the Methods section.* |
| ☒ | ☐ | A description of all covariates tested |
| ☐ | ☒ | A description of any assumptions or corrections, such as tests of normality and adjustment for multiple comparisons |
| ☐ | ☒ | A full description of the statistical parameters including central tendency (e.g. means) or other basic estimates (e.g. regression coefficient) AND variation (e.g. standard deviation) or associated estimates of uncertainty (e.g. confidence intervals) |
| ☐ | ☒ | For null hypothesis testing, the test statistic (e.g. *F*, *t*, *r*) with confidence intervals, effect sizes, degrees of freedom and *P* value noted *Give P values as exact values whenever suitable.* |
| ☒ | ☐ | For Bayesian analysis, information on the choice of priors and Markov chain Monte Carlo settings |
| ☒ | ☐ | For hierarchical and complex designs, identification of the appropriate level for tests and full reporting of outcomes |
| ☐ | ☒ | Estimates of effect sizes (e.g. Cohen's *d*, Pearson's *r*), indicating how they were calculated |

*Our web collection on statistics for biologists contains articles on many of the points above.*

## Software and code

Policy information about availability of computer code

| Data collection | Presence of phage sized DNA in agarose gels were visualized using Image Studio Lite LI-COR Biosciences V5.2. qPCR data was collected using Roche Lightcycler® 480 system II release 1.5.1.62. |
|---|---|
| Data analysis | Software used for data analysis: SeqKit (v.2), MAFFT (v7.310), trimAI (v1.4.1), RAxML (v8.2.12), iTOL, dRep (v.3.0.0), Virsorter, Vibrant, VirFinder, CheckV, R IRanges (v.2.28.0), Trimmomatic (v.0.38), Bowtie2 (v.2.3.5), Samtools (v.1.9), Deeptools (v.3.1.3), hafeZ (v1.0.2), MetaViral SPade, PROKKA (v.1.14.6), Hmmer (v.3.3.1), TABAJARA, vContact2, DGRscan, BLAST (v.2.7.1+), splicejam (v 0.0.77), fastANI (v.1.33), Primer 3, dragonfly (v.1.0.14), unicycler (v. 0.4.7), RagTag, R rstatix (v.0.7.0), R ggpubr (v.0.4.0), R gggenomes (v.0.9.9.9000), ggplot and R base stats (R version 4.1.3), bbmap (v39.06), bedtools (v.26.0), KrakenUnique (v.1.0.4), Jellyfish (v1.1.12), RapidNJ (v2.3.2), Expam (v1.2.2.5) |

For manuscripts utilizing custom algorithms or software that are central to the research but not yet described in published literature, software must be made available to editors and reviewers. We strongly encourage code deposition in a community repository (e.g. GitHub). See the Nature Portfolio guidelines for submitting code & software for further information.

## Data

Policy information about <u>availability of data</u>

All manuscripts must include a <u>data availability statement</u>. This statement should provide the following information, where applicable:

- Accession codes, unique identifiers, or web links for publicly available datasets
- A description of any restrictions on data availability
- For clinical datasets or third party data, please ensure that the statement adheres to our <u>policy</u>

All data for this study have been deposited in the European Nucleotide Archive (ENA) at EMBL-EBI under project number PRJEB64565 and accession numbers are provided in Supplementary Table S1, S3 and S12. Bacterial isolates are available through the Australian Microbiome Culture Collection (AusMiCC). Metadata of viromes used in this study along with accession numbers are provided in Supplementary Table S5. Bioinformatic scripts and Figure data is available in Figshare DOI https://doi.org/10.26180/29946902.v1

## Research involving human participants, their data, or biological material

Policy information about studies with <u>human participants or human data</u>. See also policy information about <u>sex, gender (identity/presentation), and sexual orientation</u> and <u>race, ethnicity and racism</u>.

| | |
|---|---|
| Reporting on sex and gender | n/a |
| Reporting on race, ethnicity, or other socially relevant groupings | n/a |
| Population characteristics | n/a |
| Recruitment | n/a |
| Ethics oversight | n/a |

Note that full information on the approval of the study protocol must also be provided in the manuscript.

# Field-specific reporting

Please select the one below that is the best fit for your research. If you are not sure, read the appropriate sections before making your selection.

☒ Life sciences          ☐ Behavioural & social sciences          ☐ Ecological, evolutionary & environmental sciences

For a reference copy of the document with all sections, see <u>nature.com/documents/nr-reporting-summary-flat.pdf</u>

# Life sciences study design

All studies must disclose on these points even when the disclosure is negative.

| | |
|---|---|
| Sample size | No sample size calculation was performed due to exploratory nature of project. |
| Data exclusions | Low quality reads from Illumina sequencing were excluded using Trimmomatic (SLIDINGWINDOW:4:25 MINLEN:100). Possible human, phiX-174 and cloning vector contaminants were removed from metagenomes before read mapping to temperate phage genomes. |
| Replication | Sequencing of prophage induction in pure isolate samples were performed in singlet due to cost of high throughput screens. Sequencing of synthetic microbiome was performed on 5 biological replicates. Sequencing of gene deletion mutant strain (plus wilde-type) was performed in singlet and was validated by qPCR.<br>qPCR was performed in biological triplicates with technical triplicates for each sample, no failed replications occurred. |
| Randomization | Randomization is not relevant to the microbiological experiments performed in this study. |
| Blinding | Blinding during data collection is not relevant to the microbiological experiments performed in this study. |

# Reporting for specific materials, systems and methods

We require information from authors about some types of materials, experimental systems and methods used in many studies. Here, indicate whether each material, system or method listed is relevant to your study. If you are not sure if a list item applies to your research, read the appropriate section before selecting a response.

## Materials & experimental systems

| n/a | Involved in the study |
|-----|----------------------|
| ☒ ☐ | Antibodies |
| ☐ ☒ | Eukaryotic cell lines |
| ☒ ☐ | Palaeontology and archaeology |
| ☒ ☐ | Animals and other organisms |
| ☒ ☐ | Clinical data |
| ☒ ☐ | Dual use research of concern |
| ☒ ☐ | Plants |

## Methods

| n/a | Involved in the study |
|-----|----------------------|
| ☒ ☐ | ChIP-seq |
| ☒ ☐ | Flow cytometry |
| ☒ ☐ | MRI-based neuroimaging |

## Eukaryotic cell lines

Policy information about cell lines and Sex and Gender in Research

| | |
|---|---|
| Cell line source(s) | Human colonic epithelial immortalised cells (Caco2 TC7) |
| Authentication | Genotype verified by AGRF Human Cell Line Identification Service; VIC, Australia |
| Mycoplasma contamination | Cells were routinely tested for the presence of mycoplasma contamination (MycoStrip, Invivogen, San Diego, USA) and were confirmed mycoplasma negative. |
| Commonly misidentified lines (See ICLAC register) | No commonly misidentified cell lines listed in ICLAC register was used in this study. |

## Plants

| | |
|---|---|
| Seed stocks | n/a |
| Novel plant genotypes | n/a |
| Authentication | n/a |

