## [Peer Review File · Nature]

Isolation, engineering and ecological dynamics of temperate phages from the human gut

Corresponding Author: Professor Jeremy Barr

Version 0:

Reviewer comments:

Referee #1

(Remarks to the Author)

In this manuscript, Dahlman and colleagues investigate the prevalence, diversity and activity of prophages in the human gut microbiomes. They focus on 252 diverse human gut bacterial isolates belonging to diverse phyla. The authors detected active replication of 125 prophages which, somewhat unexpectedly, corresponds to a rather modest fraction (17%) of computationally predicted prophages. The authors made a commendable effort to induce the prophages in all available strains using a panel of chemical agents/growth conditions. Interestingly, the well-known induction agents (e.g., mitomycin C) exhibited only a marginally increased induction rate compared to spontaneous induction during standard growth conditions. There is a number of other interesting observations described in the manuscript (e.g., prevalence of DGR systems). The manuscript is generally clearly written and the results, especially, the collection of active phages, will be very useful to the community for designing follow up studies, both computational and experimental, for better understanding the role of microbes and virus-host interactions in the gut. I have a few comments relating to the methodology and description of the results which I would like the authors to consider.

All but one of the active/induced prophages were assigned to the class Caudoviricetes. How many of the non-Caudoviricetes prophages were predicted computationally? Given that prophage induction has been historically nearly exclusively studied with tailed phages, could it be that the inducers/conditions used are only affecting the Caudoviricetes prophages whereas phages from other taxa are insensitive/less sensitive to them?

Related to this question, I wonder whether the authors have not eliminated the induced non-Caudoviricetes prophages due to their experimental design. Only those samples which contained the suspected "phage sized (~50 kb) DNA" (Methods, L41) were considered further and sequenced. The genomes of non-tailed phages are considerably smaller than 15 kb – 5-10 kb for filamentous phages and ~15 kb for Tectiviridae, ~10 kb for Vinavirales. Bacteroides and Prevotella are also known to carry prophages related to Microviridae (PMID: 21572966), which have genomes of <7 kb. Would all these phages, even if induced, be eliminated in your workflow?

Generally, I do not think that the overall conclusions of this study would change dramatically if the experiments were performed taking into consideration the smaller phages, because Caudoviricetes are overwhelmingly dominant in the gut. For instance, it has been estimated that Vinavirales, although present, represent only ~1% of the human gut virome (PMID: 36146653). However, the authors might want to either check their agarose gels for the presence of smaller molecular weight bands corresponding to genomes of smaller phages and eventually sequence some of them (if they were initially discarded). If this is not an option, the authors could add a note that only the dominant part of the gut phageome, i.e., tailed phages, were considered in this study.

I am puzzled by the induction of the prophage in Fusobacteria under standard growth conditions, but not in the presence of any of the inducers (Fig. 4a). In the methods, the authors write that "Fusobacteria isolate which was only sequenced in standard condition". Why? Provide an explanation in the Methods section.;

L63-64: Here or in the methods section, please provide references justifying the usage of inducing agents/conditions. For

instance, inducing potential of SCFA depletion is not a textbook knowledge.

Throughout the text, please use “prophages” rather than “prophage” when you mean plural (PMID: 21687536). Currently, the two forms are used randomly.

I was somewhat surprised not to read anything about the crassphage. None of the strains contained it?

Methods, L37: provide molecular weight of PEG used.

In the methods the authors used qPCR to enumerate the differential prophage induction, but I missed where exactly this is described in the results. From the methods section I could not figure out what was amplified with the primers Q25 and Q41.

In the Methods section, Tectaviridae should be corrected to Tectiviridae on all occasions and all virus taxon names should be written in Italics.

Referee #2

(Remarks to the Author)

COMMENTS FOR AUTHORS:

A collection of 252 diverse human gut bacteria (previously isolated) were leveraged by Dahlman et al to experimentally validate phage-host pair predictions. The authors combined in silico approaches (VirSorter, VirFinder, and Vibrant) to predict prophage signals from bacterial genomes, and in vitro experiments (induction) to determine active prophages. The authors were able to induce 17% (125 phages) of the high-quality in silico predicted prophages. These active prophages displayed distinct gene profiles compared to the non-inducible predicted prophages.

Dahlman et al., determined that Bacterioidota prophages were amongst the most prevalent in the gut virome, and that while many phages infect a single host, 7/27 Bacteroidota prophages were found to be actively replicating across bacterial species; additionally, three of the seven were found to be induced across bacterial isolates from different genera. Furthermore, the authors discovered novel phages from the Hankyphage, and posit “Hankyvirus” as a putative name for the novel genus. The authors go into some detail annotating the phage genes in silico, using Prokka and the PHROG database.

The study produced high quality data and the analysis is comprehensive and well executed. I commend the authors for their efforts here; and while I do not have major concerns about the overall manuscript and data presented (technical and biological), I do believe that additional molecular experiments are needed to validate some of their in silico insights. For example, knocking out structural and lysis-associated genes within active prophage regions and then test to see if they are still inducible. This would test the authors observation and hypothesis that “cryptic prophages encoded fewer structural and lysis-associated genes and had higher non-synonymous substitution rates in integrase and excision-related genes, providing a genetic pathway towards the inactivation and domestication of gut phages” (lines 239 – 242). Furthermore, how can the results obtained in vitro be extrapolated for what happens inside host gut environments?

MINOR COMMENTS:

Abstract:

I think the last sentence can be more impactful than stating omics is hypothesis generating and needs experimental validation. It undersells the considerable work the authors have produced. I think the implications of this study has applications in synthetic biology for biotechnological advances, and towards informing selection of probiotic strains – e.g., if prophages are predicted, are they likely to be active and cause safety/regulatory issues(?)

Line 116:

Please include summary statistics of the 1,232 metagenomic gut virome samples used in this study as supp material. A sentence (or two) explaining choice of samples (or were they randomly downloaded) would be useful. Analysis of available metadata to determine if there are any confounding variables that might explain the predominance of LoVEphage in some of these samples.

Line 160-163 re: identification of cryptic prophages by selecting “those that had been sequenced (and not induced)..”

If I understand correctly, the subset of cryptic prophages was determined by first identifying prophage signals from sequence data of bacterial isolates (prophage regions predicted using in silico tools) that weren’t induced, and then comparing to those prophages that were identified in silico AND induced in the same conditions? I think this section just needs a minor rewrite to clarify.

Line 210 and 214:

Lower case “h” for hydrogen peroxide

Figure 1:

Please include a visual legend with colours of branches on phylo tree, in addition to the text already in figure legend.

Figure 2:
Please include a visual legend explaining colours used for nodes

Figure 4a:
Please include a visual legend explaining colours used for phylum.

Figure 4c:
Increase size of visual legend for Wilby (red) and Pomma (grey); difficult to discern in print.

Figure 4 legend; line 388:
I believe "Isolate phylum show in in top bar" should read "...shown in top bar"

Extended data figure 1:
Please include a visual legend with colours of branches on phylo tree, in addition to the text already in figure legend.

Extended data figure 2:
Some of the gene segments (arrows) are small and difficult to distinguish in print and on screen. I suggest considering a different colour (other than black) to denote the DRG repeat regions on the gene maps to remove confusion with the blocks of small gene fragments. Also suggest removing asterisks to streamline graphics as two symbols to denote a single feature can be confusing. Alternatively, slimline the black block in the figure legend for DGR repeat to match that on the gene maps. The same for DGR blocks in Extended data figure 3.

Referee #3

(Remarks to the Author)

Summary of key results

This paper essentially makes the true and valid point that despite their abundance and likely importance, little is known about the dynamics of bacteriophages (prophages) within the human gut. To address this the authors characterised 125 active prophages derived from 252 diverse human gut bacterial isolates. They used seven different induction conditions and from this work were able to increase the number of experimentally validated temperate phage-host pairs within this system. They were able to induce 17% of computationally predicted prophages were induced with common induction agents. A key predictive factor in terms of whether or not the bacteriophages could be induced was if they had specific integration sites. The most prevalent bacteriophages were from Bacteroidota, prophages from this taxa had brought host ranges and polylysogeny was common.

The authors make several interesting observations from the data that they have produced, a key finding is that they show that they find diversity generating retro elements within the genomes of a subset of their prophages and they speculate that these elements have driven the evolutionary history of the prophages. They also show that the genomes of the prophages have high levels of accessory genes. The other key finding is that they show that the phages are more likely to be induced if the bacteria are polygenic and that their position within the bacterial genome can alter the induction rates. The authors have indeed done what they aimed to do which is to significantly increased the amount of bacteriophages that have been brought into culture and that this is essential in order to try to understand the functioning of bacteriophages within gut microbiomes.

Originality and significance

I agree with the authors that it is important to try to go beyond simply predicting bacteriophages using bioinformatics and I also applaud the enormous amount of work that has gone on to produce the data presented within this paper. However, I am not sure as it stands, whether this information really takes us particularly much further in our understanding of how the phages operate, than the current state-of-the-art knowledge in terms of trying to understand and unravel what the bacteriophages are doing. It is as if they have collected the tools but not really shown what and when they are active. Another problem I have with the paper as it stands is that the authors state that it is likely that most predictions within their dataset represent inactive gut prophages which they refer to as cryptic prophages - they do not justify the alternative explanation which I believe is equally likely that perhaps additional unknown induction agents may be more effective at inducing the remaining phages. In other words the conditions needed to induce the remaining bacteriophages simply have not been discovered.

Interestingly the authors have studied the phages that they define as cryptic in terms of their mutation rates and show that they have differences to those phages which do appear to be active however they don't make much of this in the discussion. I think that this paper is interesting and is original however I am not completely convinced that it is worthy of Nature due to the fact that it is really a lot of quite fascinating half stories as it were rather than one big story. However I do think that before specific studies can be done this type of study is really useful to gather the tools and in doing so to get an overall insight into the potential functioning of bacteriophages within gut microbiomes and as such the authors have taken us further to understanding what is happening. Therefore I leave it in some ways to the Editors discretion if they think this is a big enough increment – and with some revisions as suggested it could be published here.

Data and methodology

The authors state that it is likely that most predictions within our dataset represent inactive gut prophage; referred to as cryptic prophages - They do not justify the alternative explanation which I believe is equally likely that perhaps additional unknown induction agents may be more effective at inducing the remaining phages. In other words the conditions needed to induce the remaining bacteriophages simply have not been discovered.

Conclusions: robustness, validity, reliability

As stated above, I think that the authors slightly over interpret some of the data but on the whole what they have presented is robust, careful and valid.

Suggested improvements: experiments, data for possible revision

It is a particularly interesting finding that 'seven out of 27 Bacteroidota prophages were found actively replicating across bacterial species, three of which were found to be induced across bacterial isolates from different genera' however, when I went to the reference to figure 1A it was not possible for me to establish which of these genre were the ones that could be infected by the prophages I suggest that the figure is improved or there is a clear steering to supplementary data to be able to get this information.

I found the section entitled. 'Temperate phage taxonomy in inducible gut isolates' confusing despite reading it several times I was unable to fully establish if this text was following on from the paragraph before so if these were the some of the same broad host range phages that had been previously mentioned in the last paragraph.

In the section with the title 'Inducible temperate phages are prevalent within gut viromes', it was not really clear how common this bacteriophage was in comparison to other bacteriophages that are known to be widely distributed amongst human gut microbiomes such as LAK phages and CRASS phages.

Although the findings represented in the paragraph on the enhanced number of genes involved in host functions is interesting. It is quite difficult to glean the significance from the way it is written for example it is just one big paragraph with no breaks and the overall points are not clear.

References: appropriate credit to previous work?

Although it seemed that many appropriate references had been provided I felt that the reference points that I would have liked to have seen there were not always included for example IN the section with the title 'Inducible temperate phages are prevalent within gut viromes', it was not really clear how common this bacteriophage was in comparison to other bacteriophages that are known to be widely distributed amongst human gut microbiomes such as LAK phages and CRASS phages.

Clarity and context: lucidity of abstract/summary, appropriateness of abstract, introduction and conclusions

The paper is largely contextualized well and the information and the abstract introductions and conclusions acknowledges the state of play within this field.

Version 2:

Reviewer comments:

Referee #1

(Remarks to the Author)

The authors have adequately addressed all of my initial comments and, as far as I could judge, those of the other reviewers. The additional experiments performed by the authors strengthened the conclusions. I have no further comments.

Referee #2

(Remarks to the Author)

I thank the authors for a comprehensive revision of their work. I appreciate the efforts taken to address my initial concerns. The authors have conducted additional in vitro experiments to complement their already strong manuscript; including co-culture with a diverse bacterial community to better understand the role of community structure and host cell factors in phage induction.

As per my initial review, it is in my opinion that the data and methodology are valid, of high quality, and appropriate (stats). The conclusions drawn in this revised manuscript are better aligned with their findings/observations, and the writing clearly and concisely reflects this.

I do not have any major comments.

I only have very minor editorial errors.

Line 102: "taken together, 35 out of 146" instead of "taken together, 35 out 146".

Line 676: formatting - there is a square box between "150" and "L".

Referee #3

(Remarks to the Author)

I thank the authors for their full and thoughtful response to my review of their paper. I was supportive of the manuscript in principle, and had noted the originality of the study as well as the scale of the data generated, and the importance of moving beyond prophage prediction towards actually trying to understand the various functions of prophages. I did think that in its first iteration the manuscript although valuable in terms of advancing our understanding of phage dynamics in the human gut microbiome, as presented it felt like a series of tempting but partial insights rather than a single overarching discovery.

I really think that in this revision the authors have now addressed these concerns really well. In particular, I thought that the expanded the experimental framework to explore a wider range of prophage induction triggers, including bacterial community co-culture and co-culture with human epithelial cells was really useful and gave useful insights into how prophages are likely to be induced. The finding that these human-associated conditions increase prophage induction really adds biological relevance and strengthens the mechanistic aspect of this work. I think it also has relevance in other microbiomes that have been less studied but where phages may play direct or indirect roles in health status.

In my original review I also speculated that many of the prophages might not be inactive per se, but just not induced under the conditions that were tested. The authors have responded by revising their language throughout the manuscript to avoid premature labelling of prophages as 'cryptic' or 'inactive'. Instead, they now distinguish between 'inducible' and 'non-inducible' prophages, which I think is clearer and a more accurate/cautious representation of the data. I also appreciate the change to the manuscript's title to reflect this nuance.

In addition to this, the authors have added a well thought out experiment where they deleted an excision gene from an inducible prophage result and showed that it did indeed lead to loss of induction. I think this is a nice experimental validation to provide direct functional evidence for the concept of prophage domestication.

In terms of presentation, the authors have improved figure clarity (particularly Figure 1A), clarified confusing transitions in the text, and added contextual information about Crassvirales and other well-known gut phages to help benchmark their findings. I also thought the expanded discussion on mutation rates and how they relate to prophage functionality was helpful.

By also including responses to the other reviewers thoughts, I felt that the manuscript really read a lot better throughout. Anyway, in summary I think that the authors have done a really good job of addressing my original concerns and that of the other reviewers. Although the study is broad in scope the different facets are a lot more cohesive. The potential impact of these observations comes out a lot more now and I fully support the publication of this manuscript in Nature.

Referee #4

(Remarks to the Author)

In "Temperate gut phages are prevalent and diverse, yet rarely induced", Dalhman et al., take a systematic approach to study abundant, but often cryptic viral members within the human gut microbiome: temperate phages (and in particular, lysogenized prophages). Although metagenomic studies have long-been aware of their abundance, microbiome science (as well as phage biology) generally lacks foundational model systems or papers on phage-host interactions in these common mobile genetic elements in microbiomes.

Building upon a culture collection of human microbiome derived bacteria, the authors employed a comprehensive experimental approach to identify active prophages in isolate culture with known prophage inducers, in microbial community culture, in microbial community culture with human epithelial cells and isolate culture with human epithelial cells. Using a simple, but effective, bioinformatic workflow, the authors identified replicating temperate phages from these experiments. Many of the conditions more relevant to in situ growth (ie community context and epithelial cell coculture) were specifically important for prophage activation. Although the authors did not distill these differences down to specific molecules or activating pathways, the authors found more community-relevant conditions to be relevant for temperate phage induction and that temperate phages identified here are highly prevalent in virome studies comparable to crassphage. The authors further analyze notable features for prophage ecology including the prevalence of diversity generating retroelements (DGRs). They identify and test a likely genetic route for domestication through loss of integration-excision machinery. They also investigate the impact of polylysogeny on prophage induction in gut microbiome members. These analyses are well-performed and have sound conclusions.

In aggregate, this raises interest for temperate phage study within microbiome research and tractable routes to study them. I believe this paper will be seen as foundational work towards understanding commensal-phage ecology and experimental study for phages in the human gut microbiome.

Major Comment:

My primary criticism concerns clarity of communication in Fig 1, which I feel is especially important given that it is probably the biggest set of conclusions in the manuscript: the majority of bioinformatically inferred "high quality" prophages appear inactive in a pretty extensive set of conditions. This is important because the vast majority of temperate phage analysis in microbiomes are performed purely informatically and would assume they're active. The authors find more microbiome-relevant cues (inducing through community and host cell cocultures) for prophage induction beyond traditional microbiology methods (ex. addition of alkylating agents). However, navigating this as a reader was incredibly difficult in Figure 1. The conclusion Dahlman et al., are presenting involves comparisons between similar sounding experiments: (1) microbial isolate culture with a variety of inducing agents (n=252 isolates), (2) synthetic microbial community coculture (n=78 isolates), (3) synthetic community coculture with Caco2 cells (n=78 isolates) and (4) microbial isolate and Caco2 coculture (and

associated lysate, media controls) (n=32 isolates). While I support the conclusions and analyses the authors present, this took a lot of time and careful reading to understand that would be difficult for a broader readership. The authors need to more clearly delineate which subfigures refer to which set of experiments in the figure itself and not just text as it is central to both their conclusions and the quality of their work. Simple cartoons for instance would go a long way towards clarifying Fig 1.

Minor points:

- In Fig 2, please refer to the database used.
- Table TS3 has several unlabeled columns
- Since the original preprint a substantial amount of work has emerged on the DGR vignette in the paper and this section is missing a few critical citations, (below). I would recommend the authors update this section to reflect the information reported since their preprint.

(targets: <https://doi.org/10.1073/pnas.2316469121>

mechanism: <https://doi.org/10.1101/2025.03.24.644984>)

Minor points (visualization):

- Please double check that some of the figures are color-blind safe - especially those referencing microbiome composition. For instance Proteobacteria and Bacteroidota are pretty similar, especially given that several conclusions need both to be compared for conclusions. At a minimum, this impacts readability in Figs 1, 2, and 4.
- Fig 1a color scheme doesn't match the Figure legend. This color scheme also causes confusion across figures as the distinction between Bacteroides, Parabacteroides, and Bacteroidota is not maintained across figures or even within Fig 1.
- Fig 1b is missing a figure legend
- As rendered, "pink" in Fig 1f looks orange.
- In Fig2, the distinction between Crassvirales in black-boarded circles and database reference genomes is not very clear.
- In Fig2, Crassvirales is labeled on a common legend, but only refers to Fig 2b. Likely Crassvirales lies within Fig2a as well. It seems like the authors are adding this in as a response to another reviewer's comment, but is a side-analysis for this story. Even without the crassphage comparison, it is clear that some of the Bacteroidota temperate phages are among the most abundant in viromes. I would recommend the crassphage comparison be moved to the Supplement.
- In Fig 3b, it is unclear what "total" and "presence-absence" are from the figure or the figure legend. It is also unclear in the methods as well.
- I found Ext Fig 1b to be a little confusing yet was a critical figure to look at during the course of reviewing this manuscript. This is the only figure that directly compares all strains across all induction conditions. However, because not all strains were used in DMEM, lysate, and Caco2 monolayer experiments, these conditions look like "no discovery results" instead of "not tested results". Clarifying the two would be incredibly helpful as a reader.
- The figure legend for Fig 1b says "sequenced samples shown in blue". I suggest the authors reframe this to avoid ambiguity. I interpreted this section as "samples with detected phage induction shown in blue", but it's possible I was incorrect.

Please find below Dahlman et al, 2023 Referees' comments, with our responses in blue.

Referee #1 (Remarks to the Author):

In this manuscript, Dahlman and colleagues investigate the prevalence, diversity and activity of prophages in the human gut microbiomes. They focus on 252 diverse human gut bacterial isolates belonging to diverse phyla. The authors detected active replication of 125 prophages which, somewhat unexpectedly, corresponds to a rather modest fraction (17%) of computationally predicted prophages. The authors made a commendable effort to induce the prophages in all available strains using a panel of chemical agents/growth conditions. Interestingly, the well-known induction agents (e.g., mitomycin C) exhibited only a marginally increased induction rate compared to spontaneous induction during standard growth conditions. There is a number of other interesting observations described in the manuscript (e.g., prevalence of DGR systems). The manuscript is generally clearly written and the results, especially, the collection of active phages, will be very useful to the community for designing follow up studies, both computational and experimental, for better understanding the role of microbes and virus-host interactions in the gut. I have a few comments relating to the methodology and description of the results which I would like the authors to consider.

All but one of the active/induced prophages were assigned to the class Caudoviricetes. How many of the non-Caudoviricetes prophages were predicted computationally?

Running all predicted prophage genomes through a hallmark gene check for Microviridae, Tectiviridae, and Inoviridae, only five genomes had genes matching Inoviridae (no hits came back for either Microviridae or Tectiviridae). Upon closer inspection, only one of those genomes seemed a true hit for an Inovirus, as the remaining four had long genomes (>100 genes) and were either enriched for bacterial genes or encoded ParA/ParB-like partitioning systems (likely plasmids). However, as most databases (and phage prediction tools) are biased towards Caudoviricetes, we cannot exclude that non-Caudoviricetes phages are missed in these steps.

Given that prophage induction has been historically nearly exclusively studied with tailed phages, could it be that the inducers/conditions used are only affecting the Caudoviricetes prophages whereas phages from other taxa are insensitive/less sensitive to them?

We agree with the reviewer that there are inherent biases in our experimental and bioinformatic approaches towards the detection of Caudoviricetes prophages. To try and mitigate this, we applied two approaches, one based on prophage prediction, and one based solely on induction profiles (followed by viral hallmark gene check; see Extended Data Fig. 1). This second approach should detect non-Caudoviricetes prophages if they were induced, albeit the majority of induction conditions we employed are targeted towards tailed phages, as stated by the reviewer.

To address this limitation, we have now added a section in the discussion (see lines 279-281):

“There are likely biases towards induction and detection of Caudoviricetes prophages and, indeed, all but one prophage (from the Inoviridae family) belonged to this class.”

Related to this question, I wonder whether the authors have not eliminated the induced non-Caudoviricetes prophages due to their experimental design. Only those samples which contained the suspected “phage sized (~50 kb) DNA” (Methods, L41) were considered further

and sequenced. The genomes of non-tailed phages are considerably smaller than 15 kb – 5-10 kb for filamentous phages and ~15 kb for Tectiviridae, ~10 kb for Vinavirales. Bacteroides and Prevotella are also known to carry prophages related to Microviridae (PMID: 21572966), which have genomes of <7 kb. Would all these phages, even if induced, be eliminated in your workflow?

Generally, I do not think that the overall conclusions of this study would change dramatically if the experiments were performed taking into consideration the smaller phages, because Caudoviricetes are overwhelmingly dominant in the gut. For instance, it has been estimated that Vinavirales, although present, represent only ~1% of the human gut virome (PMID: 36146653). However, the authors might want to either check their agarose gels for the presence of smaller molecular weight bands corresponding to genomes of smaller phages and eventually sequence some of them (if they were initially discarded). If this is not an option, the authors could add a note that only the dominant part of the gut phageome, i.e., tailed phages, were considered in this study.

As discussed above, much of our methodology was inadvertently aimed at Caudoviricetes. We went back and checked our gels and did not observe any bright/clear bands in the ~10kb range, which, if present, should have been detected (the ladder used shows 48kb-10kb bands). Potentially phages smaller than this, for example the Microviridae as pointed out by the reviewer, could have been missed. However, any of our sequenced samples should have detected these phages if they were induced, which they were not.

To address this comment, we have added a section in the discussion (see lines 277-279):

“Caveats to our approach include experimental cut-offs for detection, and minimum amounts of DNA required for sequencing, which could exclude detection of low-level inducing prophages.”

I am puzzled by the induction of the prophage in Fusobacteria under standard growth conditions, but not in the presence of any of the inducers (Fig. 4a). In the methods, the authors write that “Fusobacteria isolate which was only sequenced in standard condition”. Why? Provide an explanation in the Methods section.

Due to limited time and resources, coupled with the fact that we only had a single Fusobacteria isolate in our collection, we decided to only sequence this isolate in standard condition (see Supplemental Method). To address this reviewers’ point, we have now sequenced this isolate in Mitomycin C and Caco2 pure culture experiments (see updated Fig. 4a).

L63-64: Here or in the methods section, please provide references justifying the usage of inducing agents/conditions. For instance, inducing potential of SCFA depletion is not a textbook knowledge.

Due to limitations on the number of references in main text, we opted to include these references in the Supplemental Method section. However, as the reviewer pointed out, depletion of SCFA is not a commonly used induction condition and most (if not all) previous studies are performed in media which do not contain SCFA. This is due to the fact that prior studies were largely limited to aerobic, non-gut bacteria where SCFA would not routinely be included in the bacterial growth media. To address this reviewer’s point, we have added explanations and references for these conditions in the main text (see lines 62-67):

“We began by exposing our isolates to seven different induction agents and conditions, which included well-known inducing agents such as Mitomycin C (0.3 and 3 µg/mL) and hydrogen peroxide (0.5 mM), along with lesser-known induction conditions with potential relevance to the gut, including the sugar substitute Stevia (3.7 and 37 mg/mL), and two starvation conditions (50% carbon depletion and 100% short-chain fatty acid (SCFA) depletion)¹⁹⁻²².”

Throughout the text, please use “prophages” rather than “prophage” when you mean plural (PMID: 21687536). Currently, the two forms are used randomly.

We have made the requested changes throughout the text.

I was somewhat surprised not to read anything about the crAssphage. None of the strains contained it?

No, none of the strains contained crAssphages. So far, only virulent CrAssvirales have been isolated, although at least one metagenomic study indicates that some crAss-like phages might be temperate (PMID: 32652061). As phage taxonomy is expanding quickly some of our phages may be included in this order in the future, even though none of the induced temperate phages were related to current CrAssvirales at the family level. For example, Phage Saffi is similar in length to crAss-like phages (>100kb) and contain some CrAssvirales like features, such as DNA and RNA polymerase.

Although none of our bacterial isolates contain crAssphages, as per the reviewers’ suggestion, we have highlighted the CrAssvirales order in our main text (see lines 144-146) and highlighted CrAssvirales genomes from *Yutin et al. 2021* in Fig. 3b to compare it to the prevalence and abundance of our inducible prophages.

Methods, L37: provide molecular weight of PEG used.

We have made the change in method section.

In the methods the authors used qPCR to enumerate the differential prophage induction, but I missed where exactly this is described in the results. From the methods section I could not figure out what was amplified with the primers Q25 and Q41.

Q25 and Q41 were the old identification numbers for phage Wilby and phage Pomma (Figure 4). We thank the reviewer for catching this error and have made changes to the methods section to correct it.

In the Methods section, Tectaviridae should be corrected to Tectiviridae on all occasions and all virus taxon names should be written in Italics.

We have made the requested changes throughout the text.

Referee #2 (Remarks to the Author):

A collection of 252 diverse human gut bacteria (previously isolated) were leveraged by Dahlman et al to experimentally validate phage-host pair predictions. The authors combined in silico approaches (VirSorter, VirFinder, and Vibrant) to predict prophage signals from bacterial genomes, and in vitro experiments (induction) to determine active prophages. The authors were able to induce 17% (125 phages) of the high-quality in silico predicted prophages. These active prophages displayed distinct gene profiles compared to the non-inducible predicted prophages.

Dahlman et al., determined that Bacteroidota prophages were amongst the most prevalent in the gut virome, and that while many phages infect a single host, 7/27 Bacteroidota prophages were found to be actively replicating across bacterial species; additionally, three of the seven were found to be induced across bacterial isolates from different genera. Furthermore, the authors discovered novel phages from the Hankyphage, and posit “Hankyvirus” as a putative name for the novel genus. The authors go into some detail annotating the phage genes in silico, using Prokka and the PHROG database.

The study produced high quality data and the analysis is comprehensive and well executed. I commend the authors for their efforts here; and while I do not have major concerns about the overall manuscript and data presented (technical and biological), I do believe that additional molecular experiments are needed to validate some of their in-silico insights. For example, knocking out structural and lysis-associated genes within active prophage regions and then test to see if they are still inducible. This would test the authors observation and hypothesis that “cryptic prophages encoded fewer structural and lysis-associated genes and had higher non-synonymous substitution rates in integrase and excision-related genes, providing a genetic pathway towards the inactivation and domestication of gut phages” (lines 239 – 242).

During the revision and resubmission process, we attempted two additional experimental approaches to strengthen the quality of our manuscript in response to reviewer suggestions. The first approach was our synthetic bacterial community to better understand the significance for the human gut environment as suggested by this reviewer below (see response below and detailed response to #Reviewer 3). This was an extensive and large experimental approach that required most of our time and effort to complete.

In addition to this, we attempted to establish genetic engineering protocols using CRISPR-Cas systems and broadly mobilised plasmids to knock-out integrase genes within prophages residing within Bacteroidota hosts (as per #Reviewer 2’s request). However, targeted genetic modification of most of these gastrointestinal bacterial species has not previously been achieved and we encountered several methodological obstacles while attempting to establish this method. Due to limited time and personnel, we eventually made the decision to focus our efforts on the synthetic bacterial community experiments and validation of human-associated induction triggers.

While we were unable to further validate these hypotheses using molecular approaches, we did increase the number of induced prophages (n=9) in our collection through human cell-bacterial co-cultures. With this increased prophage collection, we have reanalysed the frequency in gene change and dN/dS ratio (Fig. 3b & d), and found our hypotheses still hold, which provides additional validation of these mechanisms. Finally, we have added a sentence in our discussion highlighting this reviewer’s suggestion that future work should look to validate these mechanisms using molecular approaches (lines 309-311):

“Future work should incorporate molecular approaches to confirm what functional impacts these mutations have on prophage induction and replication.”

Furthermore, how can the results obtained *in vitro* be extrapolated for what happens inside host gut environments?

We thank the reviewer for raising this important point. The revised manuscript now includes further experimentation, using a diverse bacterial community co-culture, to better understand the role of community structure and host cell factors in phage induction. Here, we found that several previously non-inducible prophages were able to be induced by human-associated induction triggers providing a deeper understanding of phage induction in the host environment. We have further extrapolated upon these *in vitro* results and their implications inside host gut environments within the discussion (see lines 291-295):

“A modest increase in induction was observed within the lysed human cell products compared to cell culture media or intact cells. This is in accordance with previous observations of temperate virion expansion found in inflammatory bowel disease patients, which are associated with increased inflammation and cell death⁴.”

MINOR COMMENTS:

Abstract:

I think the last sentence can be more impactful than stating omics is hypothesis generating and needs experimental validation. It undersells the considerable work the authors have produced. I think the implications of this study has applications in synthetic biology for biotechnological advances, and towards informing selection of probiotic strains – e.g., if prophages are predicted, are they likely to be active and cause safety/regulatory issues(?)

We thank the review for this comment. As per this request, we have expanded upon this in the abstract, mentioning the impacts across fields of biotechnology, synthetic and microbiome (lines 37-41):

“More broadly, our study highlights the importance of culture-based techniques alongside experimental validation, genomics, and computational prediction. These approaches have advanced our understanding of commensal viruses within the human gut and will enable applications across synthetic biology, biotechnology, and microbiome fields.”

Line 116:

Please include summary statistics of the 1,232 metagenomic gut virome samples used in this study as supp material. A sentence (or two) explaining choice of samples (or were they randomly downloaded) would be useful.

We have added accession numbers and available metadata of gut metagenomes in the Supplementary Table 5 as well as clarified in the methods that the gut viromes were chosen based on a previous published paper (See Supplemental Materials lines 201-202; PMID: PMC8008677).

Analysis of available metadata to determine if there are any confounding variables that might explain the predominance of LoVEphage in some of these samples.

As per the reviewer’s suggestion, we performed an analysis of available metadata to look for any confounding variables with respect to LoVEphage (see analyses below). Here, we performed a count and Fisher exact test on three prevalent phage groups, including crAssvirales (see Reviewer #1 comment above), Hankyviruses (which are a prevalent temperate phage), and LoVE-like phages, and analysed their presence across geography, age, gender, and disease state. While there were significant differences across the variables tested, broadly, LoVE-like phages had fewer confounding variables than the other two prevalent viral groups. We should also note that some of the publicly available metadata was missing information for gender, disease, and geography, which may further confound these analyses. We agree with the reviewer that there is value in analysing these trends further, however we feel that this analysis is outside of the scope of our current manuscript.

To address this, we have added an additional point into our discussion (see lines 311-312) stating that future studies should investigate the abundance of these viral groups and their associations across geography, age, diet, health and disease:

“... along with investigations into the abundance and activity of important viral groups across geography, age, diet, health, and disease.”

Metadata count table:

Variable	Phage	# Viromes present
Africa	crass	26
Africa	hanky	2
Africa	LoVe	3
Europe	crass	238
Europe	hanky	8
Europe	LoVe	23
Hong Kong	crass	33
Hong Kong	hanky	8
Hong Kong	LoVe	5
North America	crass	118
North America	hanky	64
North America	LoVe	66
Adult	crass	132
Adult	hanky	38
Adult	LoVe	39
Child	crass	65
Child	hanky	35
Child	LoVe	40
Female	crass	39
Female	hanky	22
Female	LoVe	19
Male	crass	41
Male	hanky	28
Male	LoVe	14
Disease	crass	53
Disease	hanky	6
Disease	LoVe	18
Healthy	crass	127
Healthy	hanky	60
Healthy	LoVe	60

Fisher exact test, Hochberg adjusted:

		n	p	p.adj	p.adj.signif	varibale	Phage
Africa	Europe	577	8.82e-08	4.41e-07	****	Location2	crass

Africa	Hong Kong	185	0.00359	0.0108	*	Location2	crass
Africa	North America	618	1	1	ns	Location2	crass
Europe	Hong Kong	540	0.319	0.638	ns	Location2	crass
Europe	North America	973	2.1e-19	1.26e-18	****	Location2	crass
Hong Kong	North America	581	0.000194	0.000776	***	Location2	crass
Africa	Europe	577	0.446	0.569	ns	Location2	LoVe
Africa	Hong Kong	185	0.27	0.569	ns	Location2	LoVe
Africa	North America	618	0.000743	0.00371	**	Location2	LoVe
Europe	Hong Kong	540	0.569	0.569	ns	Location2	LoVe
Europe	North America	973	1.1e-05	6.6e-05	****	Location2	LoVe
Hong Kong	North America	581	0.181	0.569	ns	Location2	LoVe
Africa	Europe	577	1	1	ns	Location2	hanky
Africa	Hong Kong	185	0.0155	0.0465	*	Location2	hanky
Africa	North America	618	0.000262	0.00131	**	Location2	hanky
Europe	Hong Kong	540	0.000447	0.00179	**	Location2	hanky
Europe	North America	973	8.87e-12	5.32e-11	****	Location2	hanky
Hong Kong	North America	581	0.85	1	ns	Location2	hanky
Adult	Child	770	3.75e-40	3.75e-40	****	Age2	crass
Adult	Child	770	6.27e-05	6.27e-05	****	Age2	LoVe
Adult	Child	770	1.66e-05	1.66e-05	****	Age2	hanky
Female	Male	466	1	1	ns	Gender	crass
Female	Male	466	0.368	0.368	ns	Gender	LoVe
Female	Male	466	0.458	0.458	ns	Gender	hanky
Disease	Healthy	699	3.76e-18	3.76e-18	****	Disease2	crass
Disease	Healthy	699	0.000687	0.000687	***	Disease2	LoVe
Disease	Healthy	699	0.835	0.835	ns	Disease2	hanky

Line 160-163

Identification of cryptic prophages by selecting “those that had been sequenced (and not induced)..”. If I understand correctly, the subset of cryptic prophages was determined by first identifying prophage signals from sequence data of bacterial isolates (prophage regions predicted using in silico tools) that weren’t induced, and then comparing to those prophages that were identified in silico AND induced in the same conditions? I think this section just needs a minor rewrite to clarify.

We apologies for the lack of clarity in the sentence. Your interpretation is correct, and we have made changes to this sentence to improve its clarity (see line 196-199).

“To classify these non-induced prophages as putatively cryptic, we restricted the analysis to prophages that had been sequenced (but not induced) in the same condition(s) as their inducible counterparts, with the rational that highly similar prophages should respond to the same induction triggers.”

Line 210 and 214:

Lower case “h” for hydrogen peroxide

Changes have been made throughout the document.

Figure 1:

Please include a visual legend with colours of branches on phylo tree, in addition to the text already in figure legend.

Change made.

Figure 2:

Please include a visual legend explaining colours used for nodes

Change made.

Figure 4a:

Please include a visual legend explaining colours used for phylum.

Change made.

Figure 4c:

Increase size of visual legend for Wilby (red) and Pomma (grey); difficult to discern in print.

Change made.

Figure 4 legend; line 388:

I believe “Isolate phylum show in in top bar” should read “...shown in top bar”

Change made.

Extended data figure 1:

Please include a visual legend with colours of branches on phylo tree, in addition to the text already in figure legend.

Change made.

Extended data figure 2:

Some of the gene segments (arrows) are small and difficult to distinguish in print and on screen. I suggest considering a different colour (other than black) to denote the DRG repeat regions on the gene maps to remove confusion with the blocks of small gene fragments. Also suggest removing asterisks to streamline graphics as two symbols to denote a single feature can be confusing. Alternatively, slimline the black block in the figure legend for DGR repeat to match that on the gene maps. The same for DGR blocks in Extended data figure 3.

We have made the requested changes in figures and legends.

Referee #3 (Remarks to the Author):

Summary of key results

This paper essentially makes the true and valid point that despite their abundance and likely importance, little is known about the dynamics of bacteriophages (prophages) within the human gut. To address this the authors characterised 125 active prophages derived from 252 diverse human gut bacterial isolates. They used seven different induction conditions and from this work were able to increase the number of experimentally validated temperate phage-host pairs within this system.

They were able to induce 17% of computationally predicted prophages were induced with common induction agents. A key predictive factor in terms of whether or not the bacteriophages could be induced was if they had specific integration sites. The most prevalent bacteriophages were from Bacteroidota, prophages from this taxa had brought host ranges and polylysogeny was common.

The authors make several interesting observations from the data that they have produced, a key finding is that they show that they find diversity generating retro elements within the genomes of a subset of their prophages and they speculate that these elements have driven the evolutionary history of the prophages. They also show that the genomes of the prophages have high levels of accessory genes. The other key finding is that they show that the phages are more likely to be induced if the bacteria are polygenic and that their position within the bacterial genome can alter the induction rates.

The authors have indeed done what they aimed to do which is to significantly increased the amount of bacteriophages that have been brought into culture and that this is essential in order to try to understand the functioning of bacteriophages within gut microbiomes.

Originality and significance

I agree with the authors that it is important to try to go beyond simply predicting bacteriophages using bioinformatics and I also applaud the enormous amount of work that has gone on to produce the data presented within this paper. However, I am not sure as it stands, whether this information really takes us particularly much further in our understanding of how the phages operate, than the current state-of-the-art knowledge in terms of trying to understand and unravel what the bacteriophages are doing. It is as if they have collected the tools but not really shown what and when they are active.

We thank the reviewer for their comments and feedback on our work and the presentation of the data. We have made significant efforts to address this reviewer's concerns regarding the understanding of how phages operate in the gut. As we will outline further below, we have designed a synthetic community consisting of common human commensal bacteria, which we used to experimentally identify that human cell-associated products act as novel prophage induction triggers. We have further placed this finding within the broader context of the gut, particularly around recent observations of higher temperate phage populations associated with inflammatory bowel disease (see lines 285-295). We believe that this additional work and its implications have strengthen the impact of our work and further advanced our understanding of phages within the gut.

Another problem I have with the paper as it stands is that the authors state that it is likely that most predictions within their dataset represent inactive gut prophages which they refer to as cryptic prophages - they do not justify the alternative explanation which I believe is equally likely that perhaps additional unknown induction agents may be more effective at inducing the remaining phages. In other words the conditions needed to induce the remaining bacteriophages simply have not been discovered.

This is a valid point that we agree with and we have made several changes to our manuscript in response. Firstly, using our synthetic gut microbial community, we explore two untargeted approaches to uncover additional induction triggers. These include, bacterial community co-culture, where competition for resources, production of microbial by-products, and quorum sensing may induce prophages, along with community co-culture grown in the presence of human Caco-2 cell lines. Here, we find ~17% of prophages were induced in the bacterial community co-culture, while the addition of human cells to this community increased prophage induction to ~36%. Next, we grew 32 bacterial isolates from this community in pure culture and exposed them to Caco2 cell monolayers, Caco2 cellular lysates, or DMEM cell culture media. Through this approach, we found that 35 out of 146 prophages were inducible across all conditions (including the human cells), which represents a marginal increase to 24% of prophages being induced, compared with our initial 17% claim (for all 252 isolates). With this new data in mind, we draw the conclusion that, while community co-culture increased prophage induction, most prophage predictions are rarely induced (see lines 298-301).

Second, we have changed the language in our manuscript to reflect this reviewer's point. This begins with a change to our manuscript's title, from "*Temperate gut phages are prevalent, diverse, and predominantly inactive*" which suggests that most prophages in the gut are inactive or cryptic, to "*Temperate gut phages are prevalent and diverse, yet rarely induced*" that instead describes that most gut prophages were not experimentally inducible.

In addition, to address the reviewer's point that the conditions needed to induce the remaining prophage have not yet been discovered, we have changed our terminology from 'cryptic' or 'in-active', and now refer to prophages in our collection as being either 'inducible' or 'non-inducible'. We also limit our use of 'cryptic prophages' to those with high sequence similarity to inducible prophages, and that were confirmed as non-induced through sequencing (see Fig. 3). We believe for these 'cryptic' prophages that genetic mechanisms, rather than a lack of induction trigger, are more likely the cause of non-induction. Finally, we have expanded upon these points and the limitations of our approach in the discussion (see lines 277-283):

"Caveats to our approach include experimental cut-offs for detection, and minimum amounts of DNA required for sequencing, which could exclude detection of low-level inducing prophages. There are likely biases towards induction and detection of Caudoviricetes prophages and, indeed, all but one prophage (from the Inoviridae family) belonged to this class. Moreover, considering little is known about prophage induction triggers within the gut, it is plausible that some of our isolates carry prophages that were not induced due to a lack of appropriate induction triggers or induction may be stochastic."

Interestingly the authors have studied the phages that they define as cryptic in terms of their mutation rates and show that they have differences to those phages which do appear to be active however they don't make much of this in the discussion.

As per reviewer's request, we have expanded on these findings in the discussion (see lines 306-312):

“Moreover, non-induced predictions with high sequence similarity to experimentally induced prophages exhibited increased non-synonymous substitution rates in integrase and excision-related genes, providing a possible genetic pathway towards prophage domestication within the host genome. Future work should incorporate molecular approaches to confirm what functional impacts these mutations have on prophage induction and replication, along with investigations into the abundance and activity of important viral groups across geography, age, diet, health, and disease.”

I think that this paper is interesting and is original however I am not completely convinced that it is worthy of Nature due to the fact that it is really a lot of quite fascinating half stories as it were rather than one big story. However I do think that before specific studies can be done this type of study is really useful to gather the tools and in doing so to get an overall insight into the potential functioning of bacteriophages within gut microbiomes and as such the authors have taken us further to understanding what is happening. Therefore I leave it in some ways to the Editors discretion if they think this is a big enough increment – and with some revisions as suggested it could be published here.

We hope our additional experiments, exploration of novel induction conditions, and expanded discussion on how this knowledge impacts our understanding of bacteriophages within the gut are sufficient to convince this reviewer of the importance of this work.

Data and methodology

The authors state that it is likely that most predictions within our dataset represent inactive gut prophage; referred to as cryptic prophages - They do not justify the alternative explanation which I believe is equally likely that perhaps additional unknown induction agents may be more effective at inducing the remaining phages. In other words the conditions needed to induce the remaining bacteriophages simply have not been discovered.

Please see our responses above.

Conclusions: robustness, validity, reliability

As stated above, I think that the authors slightly over interpret some of the data but on the whole what they have presented is robust, careful and valid.

Suggested improvements: experiments, data for possible revision

It is a particularly interesting finding that ‘seven out of 27 Bacteroidota prophages were found actively replicating across bacterial species, three of which were found to be induced across bacterial isolates from different genera’ however, when I went to the reference to figure 1A it was not possible for me to establish which of these genre were the ones that could be infected by the prophages I suggest that the figure is improved or there is a clear steering to supplementary data to be able to get this information.

We have made changes in Figure 1 including a visual aid showing genus for Bacteroidota genomes. This is also shown in Extended Fig. 2 and in Supplementary Table 3.

I found the section entitled. ‘Temperate phage taxonomy in inducible gut isolates’ confusing despite reading it several times I was unable to fully establish if this text was following on from

the paragraph before so if these were the some of the same broad host range phages that had been previously mentioned in the last paragraph.

This section does not follow on directly from the broad-host range phage section preceding it. We have now rearranged these sections and added an introductory sentence here to make this distinction clearer and improved its readability (see line 122).

“We next looked to assign phage taxonomy to our induced temperate phage collection.”

In the section with the title ‘Inducible temperate phages are prevalent within gut viromes’, it was not really clear how common this bacteriophage was in comparison to other bacteriophages that are known to be widely distributed amongst human gut microbiomes such as LAK phages and CRASS phages.

In response to this reviewer’s comment, we have highlighted the Crassvirales genomes included in our database in Figure 2b to provide a reference point for the prevalence of our temperate phages within viromes. We have also stated the prevalence of the most common Crassvirales genome in the main text (line 150-151).

“Comparatively, the most abundant Crassvirales genome, belonging to the alpha/gamma family, was found in approximately 19% of the viromes investigated.”

Although the findings represented in the paragraph on the enhanced number of genes involved in host functions is interesting. It is quite difficult to glean the significance from the way it is written for example it is just one big paragraph with no breaks and the overall points are not clear.

This section has been re-written for clarity and significance.

Although it seemed that many appropriate references had been provided I felt that the reference points that I would have liked to have seen there were not always included for example IN the section with the title ‘Inducible temperate phages are prevalent within gut viromes’, it was not really clear how common this bacteriophage was in comparison to other bacteriophages that are known to be widely distributed amongst human gut microbiomes such as LAK phages and CRASS phages.

Please see our response above. We have included an additional reference for the Crassvirales order (PMID: 33594055).

The paper is largely contextualized well and the information and the abstract introductions and conclusions acknowledges the state of play within this field.

Dahlman et al, 2023 response to reviewers' comments below in blue.

Referee #1

All but one of the active/induced prophages were assigned to the class Caudoviricetes. How many of the non-Caudoviricetes prophages were predicted computationally?

Running all predicted prophage genomes through a hallmark gene check for Microviridae, Tectiviridae, and Inoviridae, only five genomes had genes matching Inoviridae (no hits came back for either Microviridae or Tectiviridae). Upon closer inspection, only one of those genomes seemed a true hit for an Inovirus, as the remaining four had long genomes (>100 genes) and were either enriched for bacterial genes or encoded ParA/ParB-like partitioning systems (likely plasmids). However, as most databases (and phage prediction tools) are biased towards Caudoviricetes, we cannot exclude that non-Caudoviricetes phages are missed in these steps.

Given that prophage induction has been historically nearly exclusively studied with tailed phages, could it be that the inducers/conditions used are only affecting the Caudoviricetes prophages whereas phages from other taxa are insensitive/less sensitive to them?

We agree with the reviewer that there are inherent biases in our experimental and bioinformatic approaches towards the detection of Caudoviricetes prophages. To try and mitigate this, we applied two approaches, one based on prophage prediction, and one based solely on induction profiles (followed by viral hallmark gene check; see Extended Data Fig. 1). This second approach should detect non-Caudoviricetes prophages if they were induced, albeit the majority of induction conditions we employed are targeted towards tailed phages, as stated by the reviewer.

To address this limitation, we have now added a section in the discussion (see lines 290-291):
“There are likely biases towards induction and detection of Caudoviricetes prophages and, indeed, all but one prophage (from the Inoviridae family) belonged to this class.”

Related to this question, I wonder whether the authors have not eliminated the induced non-Caudoviricetes prophages due to their experimental design. Only those samples which contained the suspected “phage sized (~50 kb) DNA” (Methods, L41) were considered further and sequenced. The genomes of non-tailed phages are considerably smaller than 15 kb – 5-10 kb for filamentous phages and ~15 kb for Tectiviridae, ~10 kb for Vinavirales. Bacteroides and Prevotella are also known to carry prophages related to Microviridae (PMID: 21572966), which have genomes of <7 kb. Would all these phages, even if induced, be eliminated in your workflow?

Generally, I do not think that the overall conclusions of this study would change dramatically if the experiments were performed taking into consideration the smaller phages, because Caudoviricetes are overwhelmingly dominant in the gut. For instance, it has been estimated that Vinavirales, although present, represent only ~1% of the human gut virome (PMID: 36146653). However, the authors might want to either check their agarose gels for the presence of smaller molecular weight bands corresponding to genomes of smaller phages and eventually sequence some of them (if they were initially discarded). If this is not an option,

the authors could add a note that only the dominant part of the gut phageome, i.e., tailed phages, were considered in this study.

As discussed above, much of our methodology was inadvertently aimed at Caudoviricetes. We went back and checked our gels and did not observe any bright/clear bands in the ~10kb range, which, if present, should have been detected (the ladder used shows 48kb-10kb bands). Potentially phages smaller than this, for example the Microviridae as pointed out by the reviewer, could have been missed. However, any of our sequenced samples should have detected these phages if they were induced, which they were not.

To address this comment, we have added a section in the discussion (see lines 288-290):

“Caveats to our approach include experimental cut-offs for detection, and minimum amounts of DNA required for sequencing, which could exclude detection of low-level inducing prophages.”

I am puzzled by the induction of the prophage in Fusobacteria under standard growth conditions, but not in the presence of any of the inducers (Fig. 4a). In the methods, the authors write that “Fusobacteria isolate which was only sequenced in standard condition”. Why? Provide an explanation in the Methods section.

Due to limited time and resources, and the fact that we only had a single Fusobacteria isolate in our collection, we decided to only sequence this isolate in standard media (see Supplemental Method). To address this reviewer's point, we have now sequenced this isolate in Mitomycin C and Caco2 pure culture experiments (see Fig.4 a).

L63-64: Here or in the methods section, please provide references justifying the usage of inducing agents/conditions. For instance, inducing potential of SCFA depletion is not a textbook knowledge.

Due to limitations on the number of references in main text, we opted to include these references in the Supplemental Method section. However, as the reviewer pointed out, depletion of SCFA is not a commonly used induction condition and most (if not all) previous studies are performed in media which do not contain SCFA. This is due to the fact that prior studies were largely limited to aerobic, non-gut bacteria where SCFA would not routinely be included in the bacterial growth media.

To address this reviewer's point, we have added explanations and references for these conditions in the main text (see lines 62-67):

“We began by exposing our isolates to seven different induction agents and conditions, which included well-known inducing agents such as Mitomycin C (0.3 and 3 µg/mL) and hydrogen peroxide (0.5 mM), along with lesser-known induction conditions with potential relevance to the gut, including the sugar substitute Stevia (3.7 and 37 mg/mL), and two starvation conditions (50% carbon depletion and 100% short-chain fatty acid (SCFA) depletion)¹⁹⁻²².”

Throughout the text, please use “prophages” rather than “prophage” when you mean plural (PMID: 21687536). Currently, the two forms are used randomly.

We have made the requested changes throughout the text.

I was somewhat surprised not to read anything about the crAssphage. None of the strains contained it?

No, none of the strains contained crAssphages. So far, only virulent CrAssvirales have been isolated, although at least one metagenomic study indicates that some crAss-like phages might be temperate (PMID: 32652061). As phage taxonomy is expanding some of our phages may be included in this order in the future, even though none of the induced temperate phages were related to current CrAssvirales at the family level. For example, Phage Saffi is similar in length to crAss-like phages (>100kb) and contain some CrAssvirales like features, such as DNA and RNA polymerase.

Although none of our bacterial isolates contain crAssphages, as per the reviewer’s suggestion, we have highlighted the CrAssvirales order in our main text (see lines 149-150) and highlighted CrAssvirales genomes from *Yutin et al.* 2021 in Figure 2b to compare to the prevalence and abundance of our inducible prophages.

Methods, L37: provide molecular weight of PEG used.

We have made the change in method section.

In the methods the authors used qPCR to enumerate the differential prophage induction, but I missed where exactly this is described in the results. From the methods section I could not figure out what was amplified with the primers Q25 and Q41.

Q25 and Q41 were the old identification numbers for phage Wilby and phage Pomma (Figure 4). We thank the reviewer for catching this error and have made changes to the methods section to correct it.

In the Methods section, Tectaviridae should be corrected to Tectiviridae on all occasions and all virus taxon names should be written in Italics.

We have made the requested changes throughout the text.

Referee #2

The study produced high quality data and the analysis is comprehensive and well executed. I commend the authors for their efforts here; and while I do not have major concerns about the overall manuscript and data presented (technical and biological), I do believe that additional molecular experiments are needed to validate some of their in-silico insights. For example, knocking out structural and lysis-associated genes within active prophage regions and then test to see if they are still inducible. This would test the authors observation and hypothesis that “cryptic prophages encoded fewer structural and lysis-associated genes and had higher non-synonymous substitution rates in integrase and excision-related genes, providing a genetic pathway towards the inactivation and domestication of gut phages” (lines 239 – 242).

As per the above comment, we adopted a recently published method that allows for the genetic engineering of human gut *Bacteroides* species (see Zheng *et al*⁵⁰ in Supplemental Methods). Here, we produced a gene deletion mutant of *Bacteroides faecis* strain CC01414, deleting the DNA transposition protein gene, which is required for the integration and excision of the experimentally inducible prophage Pomma (lines 224-236, Figure 3e&f and Extended Figure 4e). We confirmed this gene deletion mutant (CC01414 Φ Pomma Δ tran) using PCR and Sanger sequencing (Supplemental table 10).

We then compared the induction of the prophage Pomma between the wild-type and Δ tran strain, using both qPCR and sequencing, finding that Φ Pomma induction was completely abolished within the mutant strain. This provides functional evidence for our hypothesis that mutations within these genes may trap the prophage within the host genome (see lines 319-321):

“The deletion of one of these genes in an active prophage led to complete abolishment of induction, providing functional evidence for a genetic pathway towards prophage domestication within the host genome.”

Furthermore, how can the results obtained *in vitro* be extrapolated for what happens inside host gut environments?

We thank the reviewer for raising this important point. The revised manuscript now includes further experimentation, using a diverse bacterial community co-culture, to better understand the role of community structure and host cell factors in phage induction. Here, we found that several previously non-induced prophages were induced in a human-associated gut environment providing a deeper understanding of phage induction in relation to the human host.

We have further extrapolated upon these *in vitro* results and their implications inside host gut environments within the discussion (see lines 304-306):

“A modest increase in induction was observed within the lysed human cell products compared to cell culture media or intact cells. This is in accordance with previous observations of temperate virion expansion found in inflammatory bowel disease patients⁴.”

Abstract:

I think the last sentence can be more impactful than stating omics is hypothesis generating and needs experimental validation. It undersells the considerable work the authors have produced. I think the implications of this study has applications in synthetic biology for biotechnological advances, and towards informing selection of probiotic strains – e.g., if prophages are predicted, are they likely to be active and cause safety/regulatory issues(?)

We thank the review for this comment. As per this request, we have expanded upon this in the abstract, mentioning the impacts across fields of biotechnology, synthetic and microbiome (lines 37-41):

“More broadly, our study highlights the importance of culture-based techniques alongside experimental validation, genomics, and computational prediction. These approaches have advanced our understanding of commensal viruses within the human gut and will enable applications across synthetic biology, biotechnology, and microbiome fields.”

Due to space limitations in the abstract, we have further expanded on the impacts and applications of our study at the end of our discussions (lines 332-334):

“This will facilitate synthetic biology approaches to engineer prophages, the use of temperate virions and probiotic strains to manipulate the gut microbiome and investigate their functional impacts, and broad biotechnological applications.”

Please include summary statistics of the 1,232 metagenomic gut virome samples used in this study as supp material. A sentence (or two) explaining choice of samples (or were they randomly downloaded) would be useful. Analysis of available metadata to determine if there are any confounding variables that might explain the predominance of LoVEphage in some of these samples.

We have added accession numbers and available metadata of gut metagenomes in the Supplementary Table 5 as well as clarified in the methods that the gut viromes were chosen based on a previous published paper (See lines 800-801; PMID: PMC8008677).

As per the reviewer’s suggestion, we performed an analysis of available metadata to look for any confounding variables with respect to LoVEphage (see analyses below). Here we performed a count and Fisher exact test on three prevalent phage groups, including crAssvirales (see Reviewer #1 comment above), Hankyviruses (which are a prevalent temperate phage), and LoVE-like phages, and analysed their predominance across geography, age, gender, and disease state. While there were significant differences across the variables tested, broadly, LoVE-like phages had fewer confounding variables than the other two prevalent viral groups. We should also note that some of the publicly available metadata was missing information for gender, disease, and geography, which may further confound these analyses. We agree with the reviewer that there is value in analysing these trends further, however we feel that this analysis is outside of the scope of our current manuscript.

To address this, we have added an additional point into our discussion (see lines 292-293) stating that future studies should investigate the abundance of these viral groups and their associations across geography, age, diet, health and disease:

“Future work should investigate the abundance and activity of important viral groups across geography, age, diet, health and disease.”

Metadata count table:

Variable	Phage	# Viromes present
Africa	crass	26
Africa	hanky	2
Africa	LoVe	3
Europe	crass	238
Europe	hanky	8
Europe	LoVe	23
Hong Kong	crass	33
Hong Kong	hanky	8
Hong Kong	LoVe	5
North America	crass	118
North America	hanky	64
North America	LoVe	66
Adult	crass	132
Adult	hanky	38
Adult	LoVe	39
Child	crass	65
Child	hanky	35
Child	LoVe	40
Female	crass	39
Female	hanky	22
Female	LoVe	19
Male	crass	41
Male	hanky	28
Male	LoVe	14
Disease	crass	53
Disease	hanky	6
Disease	LoVe	18
Healthy	crass	127
Healthy	hanky	60
Healthy	LoVe	60

Fisher exact test, Hochberg adjusted:

		n	p	p.adj	p.adj.signif	varibale	Phage
Africa	Europe	577	8.82e-08	4.41e-07	****	Location2	crass
Africa	Hong Kong	185	0.00359	0.0108	*	Location2	crass
Africa	North America	618	1	1	ns	Location2	crass
Europe	Hong Kong	540	0.319	0.638	ns	Location2	crass
Europe	North America	973	2.1e-19	1.26e-18	****	Location2	crass
Hong Kong	North America	581	0.000194	0.000776	***	Location2	crass
Africa	Europe	577	0.446	0.569	ns	Location2	LoVe
Africa	Hong Kong	185	0.27	0.569	ns	Location2	LoVe
Africa	North America	618	0.000743	0.00371	**	Location2	LoVe
Europe	Hong Kong	540	0.569	0.569	ns	Location2	LoVe
Europe	North America	973	1.1e-05	6.6e-05	****	Location2	LoVe
Hong Kong	North America	581	0.181	0.569	ns	Location2	LoVe

Africa	Europe	577	1	1	ns	Location2	hanky
Africa	Hong Kong	185	0.0155	0.0465	*	Location2	hanky
Africa	North America	618	0.000262	0.00131	**	Location2	hanky
Europe	Hong Kong	540	0.000447	0.00179	**	Location2	hanky
Europe	North America	973	8.87e-12	5.32e-11	****	Location2	hanky
Hong Kong	North America	581	0.85	1	ns	Location2	hanky
Adult	Child	770	3.75e-40	3.75e-40	****	Age2	crass
Adult	Child	770	6.27e-05	6.27e-05	****	Age2	LoVe
Adult	Child	770	1.66e-05	1.66e-05	****	Age2	hanky
Female	Male	466	1	1	ns	Gender	crass
Female	Male	466	0.368	0.368	ns	Gender	LoVe
Female	Male	466	0.458	0.458	ns	Gender	hanky
Disease	Healthy	699	3.76e-18	3.76e-18	****	Disease2	crass
Disease	Healthy	699	0.000687	0.000687	***	Disease2	LoVe
Disease	Healthy	699	0.835	0.835	ns	Disease2	hanky

Line 160-163

Identification of cryptic prophages by selecting “those that had been sequenced (and not induced)..”.

If I understand correctly, the subset of cryptic prophages was determined by first identifying prophage signals from sequence data of bacterial isolates (prophage regions predicted using in silico tools) that weren’t induced, and then comparing to those prophages that were identified in silico AND induced in the same conditions? I think this section just needs a minor rewrite to clarify.

We apologies for the lack of clarity in the sentence. Your interpretation is correct and we have changed the sentence to improve clarity and readability (see line 194-197).

“To classify these non-induced prophages as putatively cryptic, we restricted the analysis to prophages that had been sequenced (but not induced) in the same condition(s) as their inducible counterparts, with the rational that highly similar prophages should respond to the same induction triggers.”

Line 210 and 214:

Lower case “h” for hydrogen peroxide

Changes have been made throughout the document.

Figure 1:

Please include a visual legend with colours of branches on phylo tree, in addition to the text already in figure legend.

Change made.

Figure 2:

Please include a visual legend explaining colours used for nodes

Change made.

Figure 4a:

Please include a visual legend explaining colours used for phylum.

Change made.

Figure 4c:

Increase size of visual legend for Wilby (red) and Pomma (grey); difficult to discern in print.

Change made.

Figure 4 legend; line 388:

I believe "Isolate phylum show in in top bar" should read "...shown in top bar"

Change made.

Extended data figure 1:

Please include a visual legend with colours of branches on phylo tree, in addition to the text already in figure legend.

Change made.

Extended data figure 2:

Some of the gene segments (arrows) are small and difficult to distinguish in print and on screen. I suggest considering a different colour (other than black) to denote the DRG repeat regions on the gene maps to remove confusion with the blocks of small gene fragments. Also suggest removing asterisks to streamline graphics as two symbols to denote a single feature can be confusing. Alternatively, streamline the black block in the figure legend for DGR repeat to match that on the gene maps. The same for DGR blocks in Extended data figure 3.

We have made the requested changes in figures and legends.

Referee #3

I agree with the authors that it is important to try to go beyond simply predicting bacteriophages using bioinformatics and I also applaud the enormous amount of work that has gone on to produce the data presented within this paper. However, I am not sure as it stands, whether this information really takes us particularly much further in our understanding of how the phages operate, than the current state-of-the-art knowledge in terms of trying to understand and unravel what the bacteriophages are doing. It is as if they have collected the tools but not really shown what and when they are active.

We thank the reviewer for their comments and feedback on our work and the presentation of the data. We have made significant efforts to address this reviewer's concern regarding the understanding of how phages operate in the gut. As we will outline further below, we have designed a synthetic community consisting of common human commensal bacteria, which we used to experimentally identify that human cell-associated products may act as novel prophage induction triggers. We have further placed this finding within the broader context of the gut, particularly around recent observations of higher temperate phage populations associated with inflammatory bowel disease (see lines 303-305). Additionally, as per Reviewer 2's request, we have engineered the gut bacterium *Bacteroides faecis* and its prophage Pomma to delete an integration and excision-related gene to provide functional data and proof of principle that non-functional mutations in these genes provides a genetic pathway for prophage domestication in the gut. We believe that this additional work and its implications have strengthened the impact of our work and further advanced our understanding of phages within the gut.

Another problem I have with the paper as it stands is that the authors state that it is likely that most predictions within their dataset represent inactive gut prophages which they refer to as cryptic prophages - they do not justify the alternative explanation which I believe is equally likely that perhaps additional unknown induction agents may be more effective at inducing the remaining phages. In other words the conditions needed to induce the remaining bacteriophages simply have not been discovered.

This is a valid point that we agree with and have made several changes to our manuscript in response. Firstly, using our synthetic gut microbial community, we explore two untargeted approaches to uncover additional induction triggers. These include, bacterial community co-culture, where competition for resources, production of microbial by-products, and quorum sensing may induce prophages, along with community co-culture grown in the presence of human Caco-2 cell lines. Here, we find ~17% of prophages were induced in the bacterial community co-culture, while the addition of human cells to this community increased prophage induction to ~35%. Next, we grew 32 bacterial isolates from this community in pure culture and exposed them to Caco2 cell monolayers, Caco2 cellular lysates, or DMEM cell culture media. Through this approach, we found that 35 out of 146 prophages were inducible across all conditions (including the human cells), which represents a marginal increase to 24% of prophages being induced, compared with our initial claim of 17% (for all 252 isolates). With this new data in mind, we draw the conclusion that, while community co-culture increased prophage induction, most prophage predictions were rarely induced.

Second, we have changed the language in our manuscript to better reflect this reviewer's point. This begins with a change to our manuscript's title, from "*Temperate gut phages are prevalent, diverse, and predominantly inactive*" which suggests that most prophages in the gut are inactive or cryptic, to "*Temperate gut phages are prevalent and diverse, yet rarely induced*" that instead describes that most gut prophages were not experimentally inducible.

In addition, to address the reviewer's point that the conditions needed to induce the remaining prophage have not yet been discovered, we have changed our terminology from 'cryptic' or 'in-active', and now refer to prophages in our collection as being either 'inducible' or 'non-inducible'. We also limit our use of 'cryptic prophages' to those with high sequence similarity to inducible prophages, and that were confirmed as non-induced through sequencing (see Fig. 3). We believe for these 'cryptic' prophages that genetic mechanisms, rather than a lack of induction trigger, are more likely the cause of non-induction. Finally, we have expanded upon these points and the limitations of our approach in the discussion (see lines 288-291 and lines 294-296):

"Caveats to our approach include experimental cut-offs for detection, and minimum amounts of DNA required for sequencing, which could exclude detection of low-level inducing prophages. There are likely biases towards induction and detection of Caudoviricetes prophages and, indeed, all but one prophage (from the Inoviridae family) belonged to this class."

"Considering little is known about prophage induction triggers within the gut, it is plausible that some of our isolates carry prophages that were not induced due to a lack of appropriate induction triggers."

Interestingly the authors have studied the phages that they define as cryptic in terms of their mutation rates and show that they have differences to those phages which do appear to be active however they don't make much of this in the discussion.

We have expanded on these findings in the discussion (see lines 317-323):

"Moreover, non-induced predictions with high sequence similarity to experimentally induced prophages exhibited increased non-synonymous substitution rates in integrase and excision-related genes. The deletion of one of these genes in an active prophage led to complete abolishment of induction, providing functional evidence for a genetic pathway towards prophage domestication within the host genome."

I think that this paper is interesting and is original however I am not completely convinced that it is worthy of Nature due to the fact that it is really a lot of quite fascinating half stories as it were rather than one big story. However I do think that before specific studies can be done this type of study is really useful to gather the tools and in doing so to get an overall insight into the potential functioning of bacteriophages within gut microbiomes and as such the authors have taken us further to understanding what is happening. Therefore I leave it in some ways to the Editors discretion if they think this is a big enough increment – and with some revisions as suggested it could be published here.

We hope our additional experiments, exploration of novel induction conditions and their biological insights, functional data on prophage domestication, and expanded discussion on

how this knowledge impacts our understanding of bacteriophages within the gut are sufficient to convince this reviewer of the importance of this work.

The authors state that it is likely that most predictions within our dataset represent inactive gut prophage; referred to as cryptic prophages - They do not justify the alternative explanation which I believe is equally likely that perhaps additional unknown induction agents may be more effective at inducing the remaining phages. In other words the conditions needed to induce the remaining bacteriophages simply have not been discovered.

Please see our responses above.

It is a particularly interesting finding that 'seven out of 27 Bacteroidota prophages were found actively replicating across bacterial species, three of which were found to be induced across bacterial isolates from different genera' however, when I went to the reference to figure 1A it was not possible for me to establish which of these genre were the ones that could be infected by the prophages I suggest that the figure is improved or there is a clear steering to supplementary data to be able to get this information.

We have made changes in Figure 1 including a visual aid showing genus for Bacteroidota genomes. This is also shown in Extended Fig. 2 and in Supplementary Table 3.

I found the section entitled. 'Temperate phage taxonomy in inducible gut isolates' confusing despite reading it several times I was unable to fully establish if this text was following on from the paragraph before so if these were the some of the same broad host range phages that had been previously mentioned in the last paragraph.

This section does not follow on directly from the broad-host range phage section preceding it. We have now rearranged the sections and added an introductory sentence here to make this distinction clearer and improved its readability (see line 121).

In the section with the title 'Inducible temperate phages are prevalent within gut viromes', it was not really clear how common this bacteriophage was in comparison to other bacteriophages that are known to be widely distributed amongst human gut microbiomes such as LAK phages and CRASS phages.

We have highlighted the Crassvirales genomes included in our database in Figure 2b to provide a reference point for the prevalence of our temperate phages within viromes. We have also stated the prevalence of the most common Crassvirales genome in the main text (line 149-150).

"Comparatively, the most abundant Crassvirales genome, belonging to the alpha/gamma family, was found in approximately 19% of the viromes investigated."

Although the findings represented in the paragraph on the enhanced number of genes involved in host functions is interesting. It is quite difficult to glean the significance from the way it is written for example it is just one big paragraph with no breaks and the overall points are not clear.

This section has been re-written for clarity and significance.

Although it seemed that many appropriate references had been provided I felt that the reference points that I would have liked to have seen there were not always included for example IN the section with the title 'Inducible temperate phages are prevalent within gut viromes', it was not really clear how common this bacteriophage was in comparison to other bacteriophages that are known to be widely distributed amongst human gut microbiomes such as LAK phages and CRASS phages.

Please see our responses to Reviewer 1 and 2 above where we have included representation and discussion of our inducible prophages in relation to crAssphage.

The paper is largely contextualized well and the information and the abstract introductions and conclusions acknowledges the state of play within this field.

Reviewer comments – Note all line numbers refer to the ‘track changes’ version of the manuscript

Referee #2 (Remarks to the Author):

I thank the authors for a comprehensive revision of their work. I appreciate the efforts taken to address my initial concerns.

The authors have conducted additional in vitro experiments to complement their already strong manuscript; including co-culture with a diverse bacterial community to better understand the role of community structure and host cell factors in phage induction.

As per my initial review, it is in my opinion that the data and methodology are valid, of high quality, and appropriate (stats). The conclusions drawn in this revised manuscript are better aligned with their findings/observations, and the writing clearly and concisely reflects this.

I do not have any major comments.

I only have very minor editorial errors.

Line 102: "taken together, 35 out of 146" instead of "taken together, 35 out 146".

Change has been made (now line 141).

Line 676: formatting - there is a square box between "150" and "L".

Change has been made (now line 972).

Referee #4

Referee #4 (Remarks to the Author):

In “Temperate gut phages are prevalent and diverse, yet rarely induced”, Dalhman et al., take a systematic approach to study abundant, but often cryptic viral members within the human gut microbiome: temperate phages (and in particular, lysogenized prophages). Although metagenomic studies have long-been aware of their abundance, microbiome science (as well as phage biology) generally lacks foundational model systems or papers on phage-host interactions in these common mobile genetic elements in microbiomes.

Building upon a culture collection of human microbiome derived bacteria, the authors employed a comprehensive experimental approach to identify active prophages in isolate culture with known prophage inducers, in microbial community culture, in microbial community culture with human epithelial cells and isolate culture with human epithelial cells. Using a simple, but effective, bioinformatic workflow, the authors identified replicating temperate phages from these experiments. Many of the conditions more relevant to in situ growth (ie community context and epithelial cell coculture) were specifically important for prophage activation. Although the authors did not distill these differences down to specific molecules or activating pathways, the authors found more community-relevant conditions to be relevant for temperate

phage induction and that temperate phages identified here are highly prevalent in virome studies comparable to crassphage. The authors further analyze notable features for prophage ecology including the prevalence of diversity generating retroelements (DGRs). They identify and test a likely genetic route for domestication through loss of integration-excision machinery. They also investigate the impact of polylysogeny on prophage induction in gut microbiome members. These analyses are well-performed and have sound conclusions.

In aggregate, this raises interest for temperate phage study within microbiome research and tractable routes to study them. I believe this paper will be seen as foundational work towards understanding commensal-phage ecology and experimental study for phages in the human gut microbiome.

Major Comment:

My primary criticism concerns clarity of communication in Fig 1, which I feel is especially important given that it is probably the biggest set of conclusions in the manuscript: the majority of bioinformatically inferred “high quality” prophages appear inactive in a pretty extensive set of conditions. This is important because the vast majority of temperate phage analysis in microbiomes are performed purely informatically and would assume they’re active. The authors find more microbiome-relevant cues (inducing through community and host cell cocultures) for prophage induction beyond traditional microbiology methods (ex. addition of alkylating agents). However, navigating this as a reader was incredibly difficult in Figure 1. The conclusion Dahlman et al., are presenting involves comparisons between similar sounding experiments: (1) microbial isolate culture with a variety of inducing agents (n=252 isolates), (2) synthetic microbial community coculture (n=78 isolates), (3) synthetic community coculture with Caco2 cells (n=78 isolates) and (4) microbial isolate and Caco2 coculture (and associated lysate, media controls) (n=32 isolates). While I support the conclusions and analyses the authors present, this took a lot of time and careful reading to understand that would be difficult for a broader readership. The authors need to more clearly delineate which subfigures refer to which set of experiments in the figure itself and not just text as it is central to both their conclusions and the quality of their work. Simple cartoons for instance would go a long way towards clarifying Fig 1.

We have added a schematic to the top of figure 1 (1a) to better visualise the different experiments conducted. We have further clarified in-text and in the figure legends which datasets related to each experiment performed.

Minor points:

- In Fig 2, please refer to the database used.

Reference to the paper in which the database was constructed has been added (see line 202).

- Table TS3 has several unlabeled columns

The missing headings have been added to TS3.

- Since the original preprint a substantial amount of work has emerged on the DGR

vignette in the paper and this section is missing a few critical citations, below). I would recommend the authors update this section to reflect the information reported since their preprint.

(targets: <https://doi.org/10.1073/pnas.2316469121>

mechanism: <https://doi.org/10.1101/2025.03.24.644984>)

We thank the reviewer for the suggested citations. We have added these citations, along with a short mention of their major findings to the associated text (see lines 233-242).

Minor points (visualization):

- Please double check that some of the figures are color-blind safe - especially those referencing microbiome composition. For instance Proteobacteria and Bacteroidota are pretty similar, especially given that several conclusions need both to be compared for conclusions. At a minimum, this impacts readability in Figs 1, 2, and 4.

The colour scheme has been changed in an attempt to improve colour blind safeness, using online resources davidmathlogic.com/colorblind and venngage.com/blog/color-blind-friendly-palette.

- Fig 1a color scheme doesn't match the Figure legend. This color scheme also causes confusion across figures as the distinction between Bacteroides, Parabacteroides, and Bacteroidota is not maintained across figures or even within Fig 1.

Colour scheme has been changed. Genus legend for Bacteroides has been kept in figure 1b as per previous comment by another reviewer.

- Fig 1b is missing a figure legend

We have added figure legend for 1c (previous 1b) and 1g (previous 1f).

- As rendered, "pink" in Fig 1f looks orange.

In improving the colour scheme towards colour blindness friendly, the pink shade has been changed in 1c and 1g.

- In Fig2, the distinction between Crassvirales in black-boarded circles and database reference genomes is not very clear.

The Crassvirales circles have been made bigger and non-translucent to better distinguish them from the other reference genomes in Extended Data Fig. 3b (see below).

- In Fig2, Crassvirales is labeled on a common legend, but only refers to Fig 2b. Likely Crassvirales lies within Fig2a as well. It seems like the authors are adding this in as a response to another reviewer's comment, but is a side-analysis for this story. Even without the crassphage comparison, it is clear that some of the Bacteroidota temperate phages are among the most abundant in viromes. I would recommend the crassphage comparison be moved to the Supplement.

We have removed the crassphage comparison from Figure 2b and instead put an extended version of Fig. 2 in Extended Data Figure 3 (a and b). In 3a, the full network of phage genomes are shown (Fig 2a is limited to network connections to prophages induced in this study) as well as highlighting Crassvirales genomes.

- In Fig 3b, it is unclear what “total” and “presence-absence” are from the figure or the figure legend. It is also unclear in the methods as well.

Clarification of these groups have been added to figure and figure legend (see line 853), as well as further elaborated on in the methods (see lines 1115-1120).

- I found Ext Fig 1b to be a little confusing yet was a critical figure to look at during the course of reviewing this manuscript. This is the only figure that directly compares all strains across all induction conditions. However, because not all strains were used in DMEM, lysate, and Caco2 monolayer experiments, these conditions look like “no discovery results” instead of “not tested results”. Clarifying the two would be incredibly helpful as a reader.

Extended Fig. 1b has been changed, removing grey background for columns DMEM, Caco2 lysate and Caco2 monolayer, similar to Fig. 4a to clarify which isolates were sequenced.

- The figure legend for Fig 1b says “sequenced samples shown in blue”. I suggest the authors reframe this to avoid ambiguity. I interpreted this section as “samples with detected phage induction shown in blue”, but it’s possible I was incorrect.

Figure legend for Extended Data Fig. 1b has been reworded for clarity, however, the blue indicate sequenced sample, not phage induction.